# Direct and indirect mortality impacts of the COVID-19 pandemic in the United States, March 1, 2020 to January 1, 2022

**Wha-Eum Lee[1], Sang Woo Park[1], Daniel M Weinberger[2], Donald Olson[3], Lone Simonsen[4], Bryan T Grenfell[1,5], Cécile Viboud[6]***

[1]Department of Ecology and Evolutionary Biology, Princeton University, Princeton, United States; [2]School of Public Health, Yale University, New Haven, United States; [3]New York City Department of Health and Mental Hygiene, New York, United States; [4]Department of Science and Environment, Roskilde University, Roskilde, Denmark; [5]Princeton School of Public Affairs, Princeton University, Princeton, United States; [6]Division of International Epidemiology and Population Studies, Fogarty International Center, National Institutes of Health, Bethesda, United States

*For correspondence:
viboudc@mail.nih.gov

**Competing interest:** The authors declare that no competing interests exist.

**Abstract** Excess mortality studies provide crucial information regarding the health burden of pandemics and other large-scale events. Here, we use time series approaches to separate the direct contribution of SARS-CoV-2 infection on mortality from the indirect consequences of the pandemic in the United States. We estimate excess deaths occurring above a seasonal baseline from March 1, 2020 to January 1, 2022, stratified by week, state, age, and underlying mortality condition (including COVID-19 and respiratory diseases; Alzheimer's disease; cancer; cerebrovascular diseases; diabetes; heart diseases; and external causes, which include suicides, opioid overdoses, and accidents). Over the study period, we estimate an excess of 1,065,200 (95% Confidence Interval (CI) 909,800–1,218,000) all-cause deaths, of which 80% are reflected in official COVID-19 statistics. State-specific excess death estimates are highly correlated with SARS-CoV-2 serology, lending support to our approach. Mortality from 7 of the 8 studied conditions rose during the pandemic, with the exception of cancer. To separate the direct mortality consequences of SARS-CoV-2 infection from the indirect effects of the pandemic, we fit generalized additive models (GAM) to age- state- and cause-specific weekly excess mortality, using covariates representing direct (COVID-19 intensity) and indirect pandemic effects (hospital intensive care unit (ICU) occupancy and measures of interventions stringency). We find that 84% (95% CI 65–94%) of all-cause excess mortality can be statistically attributed to the direct impact of SARS-CoV-2 infection. We also estimate a large direct contribution of SARS-CoV-2 infection (≥67%) on mortality from diabetes, Alzheimer's, heart diseases, and in all-cause mortality among individuals over 65 years. In contrast, indirect effects predominate in mortality from external causes and all-cause mortality among individuals under 44 years, with periods of stricter interventions associated with greater rises in mortality. Overall, on a national scale, the largest consequences of the COVID-19 pandemic are attributable to the direct impact of SARS-CoV-2 infections; yet, the secondary impacts dominate among younger age groups and in mortality from external causes. Further research on the drivers of indirect mortality is warranted as more detailed mortality data from this pandemic becomes available.

## Editor's evaluation

The authors examine the impacts of the COVID-19 pandemic on excess mortality in the US up to January 2022. The authors separate direct impacts of the pandemic from indirect impacts (disruptions), finding that most excess deaths (84%) are due to direct impacts. Moreover, in individuals

under 44 years of age, indirect effects predominate in mortality from external causes and all-cause mortality. The paper is well written and of interest to understant the impacts of the COVID-19 pandemic.

## Introduction

As the official death toll of the coronavirus disease 2019 (COVID-19) continues to grow, the full impacts of the pandemic on a range of conditions remain debated. By the end of January 2023, official statistics reported 1,056,000 deaths in the United States alone (*Johns Hopkins University, 2022*), although burden estimates that do not rely on official tallies suggest a higher death toll (*Weinberger et al., 2020*; *Karlinsky and Kobak, 2021*). A pandemic of the magnitude of COVID-19 would be expected to have secondary effects on unrelated health conditions; for instance, non-COVID-19 deaths increased in Spring of 2020 at the height of the first wave in part due to avoidance of the healthcare system (*Bollmann et al., 2020*; *Kansagra et al., 2020*; *Mafham et al., 2020*; *Woolf et al., 2020*; *Woolf et al., 2021*).

Excess mortality approaches have been used for over a century to capture the full scope of large-scale infectious disease events, heatwaves, and earthquakes, by measuring the rise in mortality over a historical baseline (*Serfling, 1963*; *Weinberger et al., 2020*). In the early phase of the pandemic, these approaches highlighted substantial underestimation in official statistics of COVID-19 deaths due to limited viral testing (*Weinberger et al., 2020*; *Kobak, 2021*). More recent analyses have examined excess mortality patterns for specific causes of death by age and socio-demographic groups and have compared the COVID-19 death toll between countries (*Banerjee et al., 2020*; *Islam et al., 2021*; *Karlinsky and Kobak, 2021*; *Mena et al., 2021*; *Rossen et al., 2021*; *Woolf et al., 2021*; *COVID-19 Excess Mortality Collaborators, 2022*). Yet, separating the direct impact of SARS-CoV-2 infection on mortality from the other consequences of the pandemic remains challenging.

To measure the direct and indirect effects of the COVID-19 pandemic, it is important to enumerate the mechanisms that could generate these effects and understand how they would manifest in mortality statistics. We first consider the direct effects of COVID-19 as those deaths that resulted from SARS-CoV-2 infection and its complications. When there is evidence of SARS-CoV-2 infection in the days or weeks before death, either virologically or clinically, it is likely that these deaths will receive a COVID-19 code during certification and these deaths will appear in official statistics. We would, however, expect variation between states in the death certification process for COVID-19. Additionally, a number of deaths triggered by SARS-CoV-2 infection could result from a complicated and protracted pathologic process, especially in patients with multiple underlying chronic conditions, who may lack a history of SARS-CoV-2 testing, and whose death may not be ascribed to COVID-19. For instance, a death in a diabetic patient could have been triggered by an undetected SARS-CoV-2 infection, resulting in a primary code of diabetes, with a SARS-CoV-2 code either lacking or listed as contributing condition. We would then expect a rise in diabetes mortality to coincide with a rise in COVID-19 cases. A similar phenomenon has been reported for influenza, with mortality from chronic conditions rising concomitantly with influenza-associated respiratory mortality in epidemic and pandemic seasons (*Reichert et al., 2004*; *Quandelacy et al., 2014*).

In addition to the direct impacts of COVID-19, there will be positive and negative changes in mortality during the pandemic period that are not associated with SARS-CoV-2 infection and its complications. We refer to these changes as indirect impacts. Reasons for these changes include avoidance of the healthcare system for treatment of acute conditions and for management of underlying chronic conditions, stressed healthcare systems in a period of high COVID-19 incidence, mental health issues in families of patients severely affected by COVID-19, societal disruptions (*Sharma et al., 2021*), decreased social interaction that depresses circulation of endemic pathogens, and decreased air pollution. The indirect impacts of the pandemic on mental health, violence, and addiction remain particularly debated, with potentially large impacts on mortality (*Faust et al., 2021b*; *Faust et al., 2021c*). These indirect mortality changes may or may not coincide temporally with COVID-19 waves.

In the United States, there was substantial geographic and temporal heterogeneity in the trajectory of the COVID-19 pandemic, along with differences in the strength and types of interventions implemented to mitigate COVID-19. In a large country with standardized death ascertainment like the United States, these heterogeneities provide an opportunity to separate the contributions of viral

infection from other drivers of mortality. Here, we apply time series approaches to four large waves of COVID-19 from March 1, 2020 to January 1, 2022, to separate the direct consequences of SARS-CoV-2 infection on age- state- and cause-specific mortality from the indirect consequences associated with hospital strain and interventions. Our analyses indicate that the direct and indirect effects of the pandemic vary substantially by chronic condition and age group. A better understanding of these effects is particularly important for the mitigation of future large-scale pandemics.

## Results

### Overall mortality patterns

We compiled weekly US mortality data by age and state, from August 1, 2014 to January 1, 2022 for eight underlying conditions (all causes, respiratory conditions, Alzheimer's disease, cancer, cerebrovascular diseases, diabetes, heart diseases, external causes; see supplement for disease codes and *Centers for Disease Control and Prevention, 2022a*; *Centers for Disease Control and Prevention, 2022b*). External causes include suicides, accidents, homicides, and poisoning from opioids and other substances, among other conditions. Respiratory mortality includes deaths ascribed to COVID-19 (ICD code U07), influenza, pneumonia, and chronic lower respiratory diseases. This is our most specific indicator of excess deaths directly attributable to SARS-CoV-2 infection. We used the data until March 1, 2020 to calibrate seasonal regression models and project expected mortality baselines in the absence of a pandemic (See methods for details). Models were adjusted for influenza circulation. Pandemic excess mortality was the difference between observed and expected baseline mortality from March 1, 2020 to January 1, 2022 (see https://github.com/viboudc/DirectIndirectCOVID19MortalityEstimation; *Viboud, 2023* for data and code).

Across the United States from March 1, 2020 to January 1, 2022, there were 848,866 cumulative deaths officially attributed to COVID-19, namely, with COVID-19 as the underlying cause of death. During the same period, we estimate 757,600 (95% Confidence Intervals (CI) 725,200–788,100) excess respiratory deaths and 1,065,200 (95% CI 909,800–1,218,000) excess deaths due to all-cause (*Table 1*). National mortality patterns comprise four waves from March 1 to June 20, 2020 (wild-type variant); June 21 to September 19, 2020 (wild-type variant); September 20, 2020 to June 19, 2021 (wild-type and Alpha variants); June 20 to November 11, 2021 (Delta variant). A recrudescence of mortality in the last weeks of 2021 was attributable to the co-circulation of the Delta and Omicron variants. The timing and intensity of mortality varied greatly by state (*Figure 1A* and *Appendix 1—figures 1–8*). The first wave was concentrated in Northeastern states, while Southern and Western states experienced mortality increases during later waves. A sensitivity analysis based on the length of the historic data used for calibration of the model baseline is shown in *Appendix 1—figure 9*. All-cause and respiratory disease estimates, as well as national estimates, were particularly robust to the choice of the calibration period.

Next, to validate our excess mortality approach, we compared our estimates with serology (see methods for details). Excess respiratory mortality showed a significant, positive correlation with CDC seroprevalence surveys (*Centers for Disease Control and Prevention, 2022g*) conducted in late December 2021 in each state (*Figure 1B*). Seroprevalence estimates ranged between 11.1 and 47.7% across states, with a population-weighted national seroprevalence of 34.6%. New York and Alabama experienced higher than predicted excess mortality with respect to their reported serologic infection rates, while Illinois and Michigan had the reverse pattern. The nationwide infection fatality rate (IFR) was estimated at 0.67% (95% CI 0.60–0.73%) based on excess respiratory mortality and 0.89% (95% CI 0.77–1.02%) based on all-cause excess mortality (*Appendix 1—figure 10*). Sensitivity analyses based on the maximum reported seroprevalence at any time point of the study period indicate that New York remained an outlier, with Illinois and Texas showing the reverse pattern (*Appendix 1—figure 10*). Use of official COVID-19 deaths determined an IFR of 0.72% (95%CI 0.62–0.81%); interestingly, serology was more highly correlated with excess respiratory deaths than with official COVID-19 deaths (*Appendix 1—figure 10*). The IFR was significantly higher in individuals over 65 years, estimated at 5.5% (95% CI 4.5–6.6%) based on all-cause excess respiratory mortality.

Next, we compared the mortality burden of COVID-19 and influenza. We estimated excess mortality for the severe November 2017 to March 2018 influenza A/H3N2 season and for the large wave of COVID-19 in November 2020 to March 2021 (see Appendix for details). We find that nationally, over

Table 1. Reported COVID-19 deaths by US jurisdiction, Compared with Excess Deaths from All-Causes and Respiratory Diseases: March 1, 2020 to January 1, 2022.

| Jurisdiction | Estimated excess all-cause deaths per 100,000, (95% prediction interval) | No. estimated excess all-cause deaths (95% prediction interval) | No. estimated excess respiratory deaths (95% prediction interval) | No. reported COVID-19 deaths* | Ratio of COVID-19 deaths to all-cause excess deaths | Ratio of COVID-19 deaths to respiratory excess deaths |
|---|---|---|---|---|---|---|
| United States | 318 (272–364) | 1,065,200 (909,800–1,218,000) | 757,600 (725,200–788,100) | 848,886 | 0.80 | 1.12 |
| Alabama | 569 (406–727) | 26,900 (19,200–34,400) | 15,000 (13,000–16,800) | 16,425 | 0.61 | 1.09 |
| Arizona | 414 (328–498) | 35,000 (27,700–42,100) | 22,900 (21,000–24,600) | 23,381 | 0.67 | 1.02 |
| Arkansas | 450 (283–612) | 13,800 (8700–18,700) | NA | 9363 | 0.68 | NA |
| California | 286 (234–336) | 120,500 (98,700–141,800) | 75,500 (69,800–80,700) | 81,910 | 0.68 | 1.08 |
| Colorado | 284 (173–393) | 15,000 (9100–20,800) | NA | 11,280 | 0.75 | NA |
| Connecticut | 238 (98–374) | 8700 (3600–13,700) | NA | 9451 | 1.08 | NA |
| Florida | 342 (279–404) | 80,200 (65,400–94,600) | 54,900 (51,400–58,200) | 60,704 | 0.76 | 1.11 |
| Georgia | 366 (287–443) | 39,600 (31,100–48,000) | 25,400 (23,100–27,600) | 27,763 | 0.70 | 1.09 |
| Illinois | 269 (198–339) | 35,600 (26,200–44,800) | 25,300 (22,800–27,600) | 28,509 | 0.80 | 1.13 |
| Indiana | 325 (213–436) | 21,600 (14,100–28,900) | 16,400 (14,100–18,500) | 19,830 | 0.92 | 1.21 |
| Iowa | 195 (26–361) | 5900 (800–10,900) | NA | 8295 | 1.41 | NA |
| Kansas | 241 (75–401) | 7000 (2200–11,600) | NA | 7316 | 1.05 | NA |
| Kentucky | 421 (278–560) | 18,600 (12,300–24,800) | NA | 13,313 | 0.71 | NA |
| Louisiana | 451 (317–581) | 21,300 (15,000–27,400) | NA | 13,984 | 0.66 | NA |
| Maryland | 262 (163–360) | 17,000 (10,600–23,400) | NA | 12,832 | 0.75 | NA |
| Massachusetts | 196 (95–297) | 13,500 (6500–20,300) | NA | 15,799 | 1.17 | NA |
| Michigan | 251 (165–336) | 26,800 (17,600–35,900) | 22,200 (19,600–24,700) | 27,287 | 1.02 | 1.23 |
| Minnesota | 149 (42–253) | 8800 (2500–14,900) | NA | 11,015 | 1.25 | NA |
| Mississippi | 477 (305–645) | 14,500 (9300–19,600) | NA | 11,069 | 0.76 | NA |
| Missouri | 309 (190–426) | 19,200 (11,800–26,400) | 15,400 (13,300–17,200) | 17,005 | 0.89 | 1.11 |
| Nevada | 348 (211–479) | 12,000 (7300–16,500) | NA | 9172 | 0.76 | NA |
| New Jersey | 320 (238–401) | 30,300 (22,500–38,000) | 25,500 (23,600–27,300) | 27,770 | 0.92 | 1.09 |
| New York | 353 (287–416) | 69,000 (56,300–81,500) | 56,700 (48,400–63,600) | 62,339 | 0.90 | 1.10 |
| Ohio | 400 (296–502) | 46,600 (34,500–58,500) | 30,900 (27,800–33,700) | 35,633 | 0.77 | 1.15 |
| Oklahoma | 416 (263–566) | 15,600 (9800–21,100) | NA | 13,098 | 0.84 | NA |
| Oregon | 209 (76–338) | 8900 (3200–14,400) | NA | 5801 | 0.65 | NA |
| Pennsylvania | 347 (259–434) | 44,400 (33,100–55,500) | 34,500 (31,800–37,000) | 38,954 | 0.88 | 1.13 |
| South Carolina | 437 (301–570) | 21,100 (14,500–27,500) | NA | 15,202 | 0.72 | NA |
| Tennessee | 411 (292–526) | 27,800 (19,800–35,700) | 20,200 (18,000–22,200) | 21,580 | 0.78 | 1.07 |
| Texas | 364 (311–417) | 104,300 (89,000–119,400) | 76,000 (72,200–79,500) | 82,328 | 0.79 | 1.08 |
| Virginia | 236 (149–320) | 21,000 (13,300–28,500) | 12,800 (10,700–14,800) | 15,824 | 0.75 | 1.22 |
| Washington | 172 (77–264) | 12,800 (5700–19,600) | NA | 9868 | 0.77 | NA |
| Wisconsin | 219 (−23–43) | 13,100 (−1400–26,600) | NA | 12,362 | 0.94 | NA |

*As reported by National Center for Health Statistics. States are ordered alphabetically. No. of reported COVID –19 deaths (any death with COVID-19 as underlying cause) until December 31, 2021 as available on June 14, 2022, were obtained from the NCHS website (**Centers for Disease Control and Prevention, 2022d**).

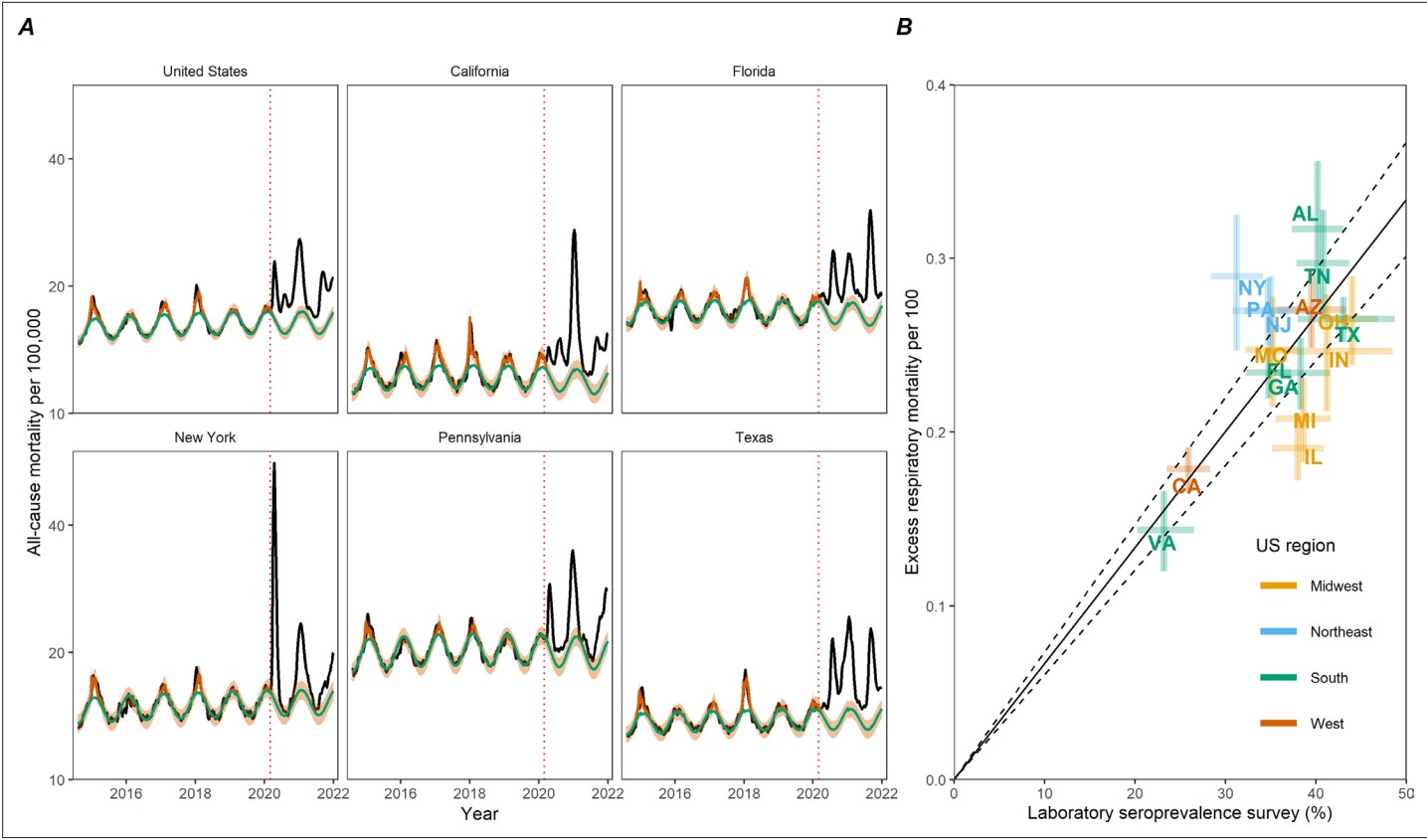

**Figure 1.** Weekly mortality rates (per 100,000) for select US jurisdictions and validation of COVID-19 excess mortality estimates against serology. (**A**) Weekly all-cause mortality rate per 100,000 in the United States and top five most populated states, August 2, 2014 to January 1, 2022. Black lines show observed data. Green line shows the seasonal model baseline. The red solid line shows the seasonal variation accounting for influenza circulation. The orange shaded areas show the upper and lower 95% confidence intervals (CIs). The dotted vertical red line marks the start of the pandemic on March 1, 2020. (**B**) Comparison between estimated excess respiratory mortality rates and cumulative COVID-19 seroprevalence estimates from the Centers for Disease Control and Prevention (CDC) as of December 31, 2021. Each point corresponds to a state; observations are shown for 16 states which have enough resolution in respiratory mortality data. Error bars represent 95% CIs on serology and excess mortality estimates. The black line and dotted region represent a linear regression fit and the associated 95% CI for a model without intercept.

this 5 month period, the mortality burden of COVID-19 was 5.7-fold higher than that of influenza based on all-cause excess mortality. A similar pattern was seen in all states median ratio of COVID-19 to influenza excess mortality rates across states, 5.8 (IQR, 5.0–7.8), (*Appendix 1—figure 11*).

## Direct and indirect pandemic impacts by cause of death

To probe the direct and indirect mortality impacts of the pandemic, we assessed whether the trajectories of various mortality categories were synchronous with that of respiratory mortality. Synchronicity would signal a direct impact of SARS-CoV-2 infection on these mortality categories. Overall, during the March 1, 2020 to January 1, 2022 pandemic period, excess mortality increased for 6 of the 7 nonrespiratory conditions studied, although the timing and intensity of the rise varied by disease (*Table 2*, *Figure 2*). Cancer was the only mortality condition that did not increase during the pandemic. Cancer deaths have remained below historic levels since March 2020, although cumulative departures from baseline were not significantly different from zero (*Table 2*). In contrast, mortality from chronic conditions such as Alzheimer's, diabetes, and heart disease rose during the pandemic, with the trajectory of excess mortality matching the pattern of respiratory mortality in the 4 pandemic waves (*Figure 2* for national patterns, and *Appendix 1—figures 3–8* for state-specific data). Across these causes of death, the first excess mortality peaks occurred within one week of the first respiratory mortality peak on April 18, 2020, with the most pronounced synchronicity patterns observed in the first wave in New York and New Jersey. Across chronic conditions, the peak of excess mortality was highest during the winter of 2020 to 21, or the third wave of the pandemic (*Appendix 1—figure 12*).

**Table 2.** Estimation of the direct impacts of COVID-19 on non-respiratory conditions.

| Cause of Death | No. estimated excess deaths (95% prediction interval) | % of excess deaths directly attributable to COVID-19 (95% prediction interval)* |
|---|---|---|
| All-cause | 1,065,200 (909,800–121,8000) | 84% (65, 94) |
| Alzheimer's | 25,300 (12,600–37,600) | 70% (45, 89) |
| Diabetes | 24,700 (15,900–33,300) | 70% (45, 93) |
| Heart diseases | 51,300 (7,400–94,300) | 73% (32, 94) |
| Cerebrovascular diseases | 16,600 (5,300–27,800) | 26% (−17, 62) |
| External causes | 102,800 (81,400–123,700) | −48% (−64, −23)† |
| Cancer | 4,300 (−18,100–26,500) | N/A‡ |

* Regression estimates of the direct impact of COVID-19 on cause-specific excess mortality, where weekly cause-specific excess mortality is regressed against COVID-19 intensity, strength of interventions, and ICU occupancy, using gam models. Estimates are based on comparison of predictions from the full model with counterfactual predictions where the COVID-19 term is set to zero.

†COVID-19 intensity is significant but negatively associated with excess mortality from external causes, hence the estimated attributable fraction is negative.

‡COVID-19 intensity is not retained in the cancer model.

We found a significant rise in deaths from external causes during the pandemic period from March 1, 2020 to January 1, 2022, corresponding to 102,800 (95% CI 81,400–123,700) cumulative excess deaths nationally (*Figure 2* and *Table 2*). The largest excess mortality rates from external causes were found in states that also had high baseline death rates from these conditions (*Appendix 1—figure 13*). However, the weekly trajectory of mortality from external causes did not align with that of respiratory mortality. We further analyzed subcategories of external causes that were available on a monthly resolution (see *Figure 3* and methods for data). The largest excess death tolls observed during this period were from accidents and injuries (43,600 excess deaths (95% CI 17,200–70,000), a 12% increase over baseline), drug overdoses (25,300 deaths (95% CI 12,000–38,700), 16% increase), and assaults and homicides (8,000 deaths (95% CI 3,700–12,200), 20% increase, *Table 3*). Overdoses were the first to peak in May 2020, followed by accidents and assaults in July 2020. Notably, mortality from suicides remained at historic levels throughout the end of the study period.

We saw evidence of increased synchronicity in multiple causes of death during the pandemic, which is a signature of the direct effects of SARS-CoV-2 infection on mortality. During the period March 1, 2020 to January 1, 2022, and compared to historical patterns, all-cause mortality became more correlated with excess deaths from respiratory conditions in all 16 states with available respiratory estimates (*Appendix 1—figure 14* and methods for details). States that experienced high cumulative excess respiratory deaths had concomitantly high excess mortality from all-causes (Spearman rho=0.81, 95% CI: 0.48–0.94), attesting to the large impact of COVID-19 on total mortality (*Appendix 1—figure 15*). Synchrony between excess deaths from underlying respiratory diseases and excess deaths from underlying chronic conditions increased during the pandemic in a subset of states (*Appendix 1—figure 14*), particularly for diabetes (n=8 states), Alzheimer's (n=5), heart diseases (n=4), and cerebrovascular diseases (n=4). In contrast, excess deaths recorded as due to cancer or external causes showed either no change (in most states) or declining synchrony (in one or two states) with respiratory mortality during the pandemic.

Next, to quantify the direct and indirect impacts of the pandemic on different causes of death, we used GAM to regress weekly cause-specific excess mortality on official COVID-19 deaths, the strength of non-pharmaceutical interventions, and hospital ICU occupancy (see methods for statistical approach, and *Figure 4* and *Appendix 1—figure 16* for results). We used official COVID-19 deaths as a proxy for the direct impact of SARS-CoV-2 infection on mortality. The variables measuring the strength of interventions (Oxford contingency index, *Oxford University, 2021*) and ICU occupancy (*Health and Human Services, 2022*) allowed for the estimation of the indirect consequences of the pandemic on mortality. We found a major direct impact of SARS-CoV-2 infection on mortality from all-cause, diabetes, heart disease, cerebrovascular diseases, and Alzheimer's; namely, the strongest

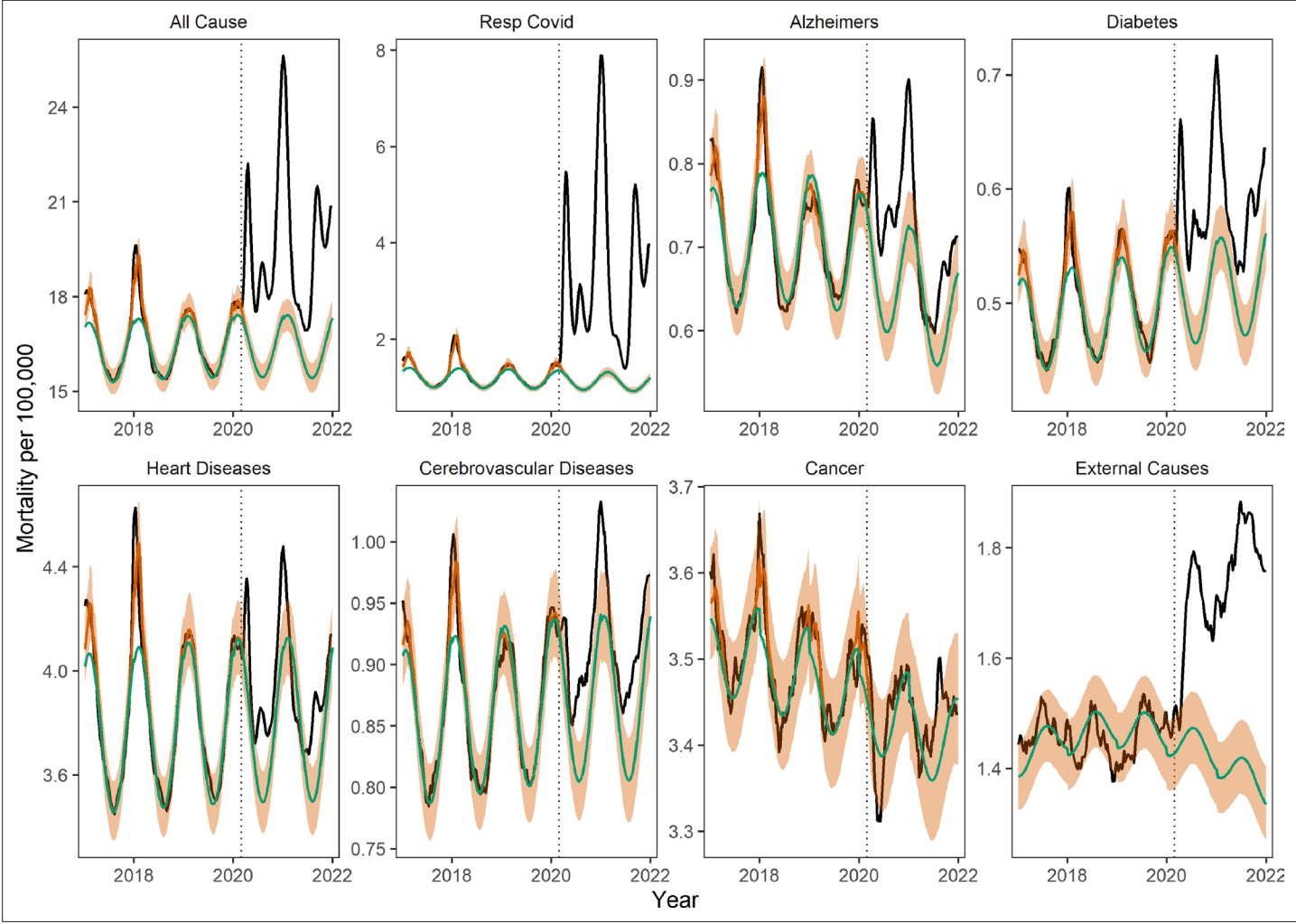

**Figure 2.** Weekly national mortality rates and model baselines (per 100,000) for eight causes of death. The black line shows observed data, the green line shows the seasonal model baseline, the orange shaded areas the 95% Confidence Interval (CI) on the seasonal baseline, and the red line shows model predictions with seasonal variation and influenza circulation. Excess mortality attributed to the COVID-19 pandemic is defined as the area between the black and green line from March 1, 2020 onwards. The dotted black vertical line marks the start of the pandemic on March 1, 2020.

predictor of excess mortality from these causes was COVID-19 deaths. The relationship between these mortality conditions and COVID-19 deaths, while non-linear, was typically monotonically increasing. Non-pharmaceutical interventions and ICU occupancy variables were also statistically associated with excess mortality, although the form of the relationship was more complex. Non-pharmaceutical intervention variables had a curvilinear relationship with excess deaths, consistent with different mechanisms affecting different periods of the pandemic. At lower levels of interventions (measured by the Oxford contingency index between 0 and 50), representing the early stages of the lockdown in March 2020, excess mortality rose with interventions. Later in the pandemic, increased interventions were estimated to have a beneficial effect on excess mortality, driven by comparison between late 2020 when interventions were strengthened in response to increasing COVID-19 activity (Oxford index above 60), and Spring 2021 when interventions were relaxed (Oxford index between 50 and 60). The relationship with the ICU occupancy variable was more difficult to interpret, varied between causes of death, and had the lowest statistical significance of the three variables tested. Furthermore, all mortality conditions were not equally well captured by our models: the best model fit was for all-cause mortality (R2=96%) and the worst was for cerebrovascular diseases (R2=47%; *Figure 4*).

On a national level, the GAM approach estimated that 84% (95% CI 65–94%) of all-cause excess deaths were attributable to the direct impact of SARS-CoV-2 infection, while the proportion was 73% (95% CI, 32–94%) for heart diseases, 70% (95% CI: 45–89%) for Alzheimer's, and 70% (95% CI:

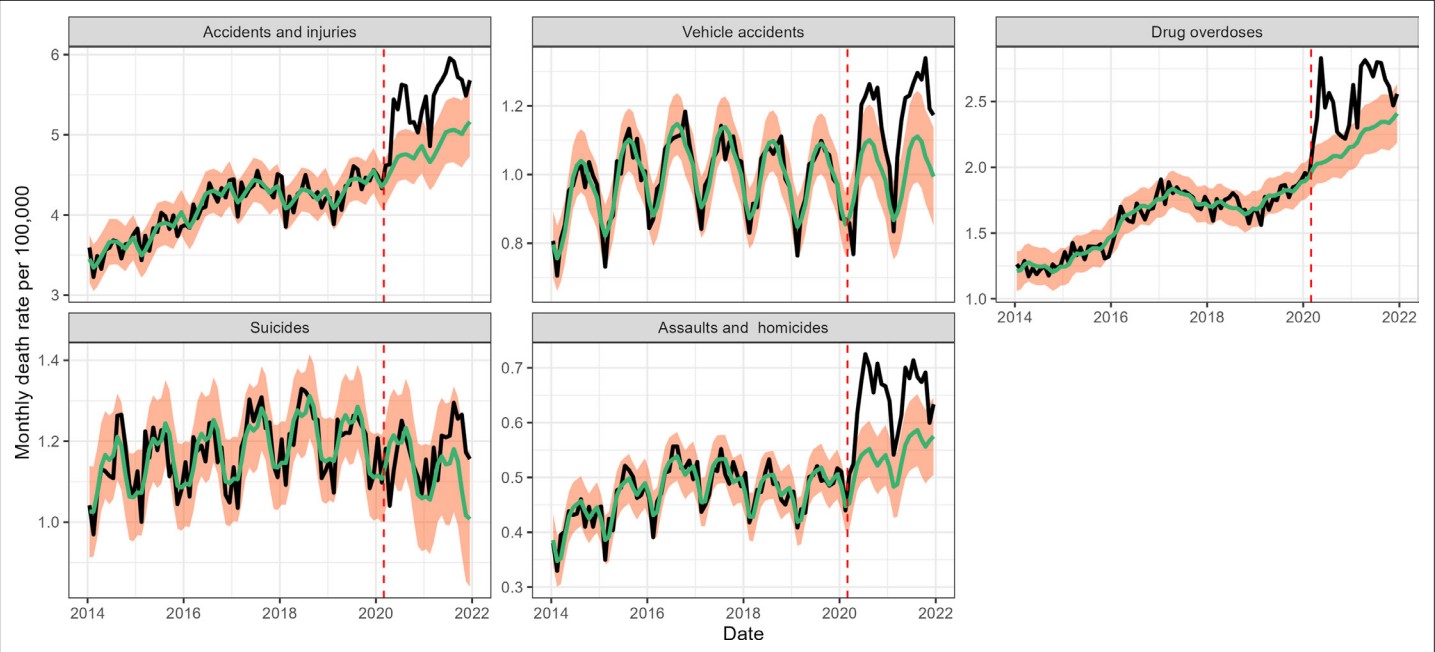

**Figure 3.** Monthly national deaths by subcategory of external causes of death from January 2014 to December 2021. The black line shows observed data, the green line shows the seasonal model baseline, and the orange shading represents the 95% Confidence Interval (CI) on the seasonal baseline. The dotted red vertical line marks the start of the pandemic period of excess mortality on March 1, 2020.

45–93%) for diabetes (*Table 2*). The contribution of COVID-19 on cerebrovascular diseases was not statistically significant.

Applying a similar GAM approach to excess mortality from cancer and external causes revealed that these conditions were more strongly associated with the intervention and ICU occupancy variables than with COVID-19. Stricter interventions were associated with a nearly linear increase in external cause mortality and a decline in cancer mortality. COVID-19 deaths had a negative effect on excess mortality from external causes (i.e. high COVID-19 activity coincided with fewer excess deaths from external causes, *Table 2*). The model for cancer had the worst fit of all conditions studied, while the model for external causes had an intermediate fit (R2=18% vs 58% respectively).

State-level analyses yielded similar estimates of direct and indirect pandemic effects as in national analyses (*Appendix 1—figure 17*). The median proportion of all-cause excess deaths attributed to direct COVID-19 effects was 81% by the GAM approach (inter-quartile range across states, 63–90%, *Appendix 1—figure 17*). State-level analyses confirmed the direct impact of COVID-19 on Alzheimer's,

**Table 3.** Excess mortality for different subcategories of external deaths during the COVID-19 pandemic period, March 2020 to December 2021.

Estimates are based on a seasonal regression model fitted to monthly data (as shown in *Figure 3*).

| Underlying cause of death | No of excess deaths (95% prediction intervals) | Ratio of excess deaths to baseline deaths (95% confidence intervals)* |
|---|---|---|
| Accidents (unintentional injuries) | 43,600 (17,200–70,000) | 0.12 (0.05–0.2) |
| Motor vehicle accidents† | 9,600 (1,000–18,200) | 0.13 (0.01–0.24) |
| Drug overdoses | 25,300 (12,000–38,700) | 0.16 (0.07–0.24) |
| Assaults and homicides | 8,000 (3,700–12,200) | 0.2 (0.09–0.31) |
| Suicides | 3,000 (−7,000–13,100) | 0.04 (−0.08–0.16) |

*This should be interpreted as the percent increase over baseline. For instance, mortality from accidents increased by 12% (95% CI, 5–20%) during the period March 2020 to December 2021 (p<0.05), relative to baseline pre-pandemic levels.

†Motor vehicle accidents are a subcategory of accidents.

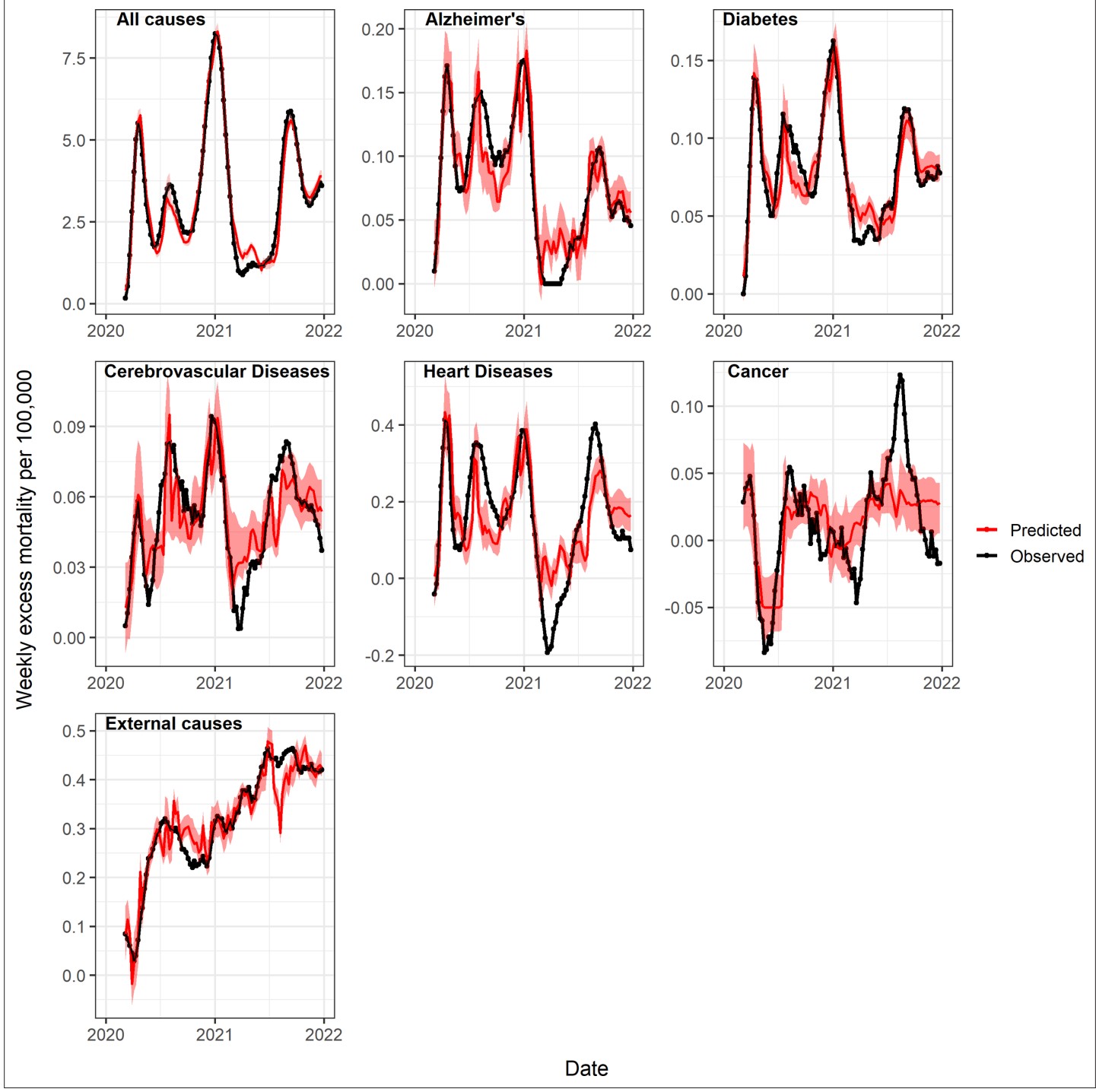

**Figure 4.** Observed and predicted excess death rates by condition, United States, March 1, 2020 to January 1, 2022, using generalized additive models (GAM) with weekly COVID-19 deaths, intensive care unit (ICU) occupancy, and a proxy for the strength of interventions as covariates. Observed values are in black and predicted values are in red (mean=dark red, 95% Confidence Interval (CI) in lighter red). See also *Appendix 1—figure 16* for a comparison of predicted and observed values, and *Appendix 1—figures 18 and 19* for age-specific models.

diabetes, and heart diseases, although the effect size was generally attenuated compared to national analyses. Consistent with national analyses, the effect of COVID-19 on mortality from cerebrovascular disease and cancer was low or non-significant, while COVID-19 had a negative effect on mortality from external causes.

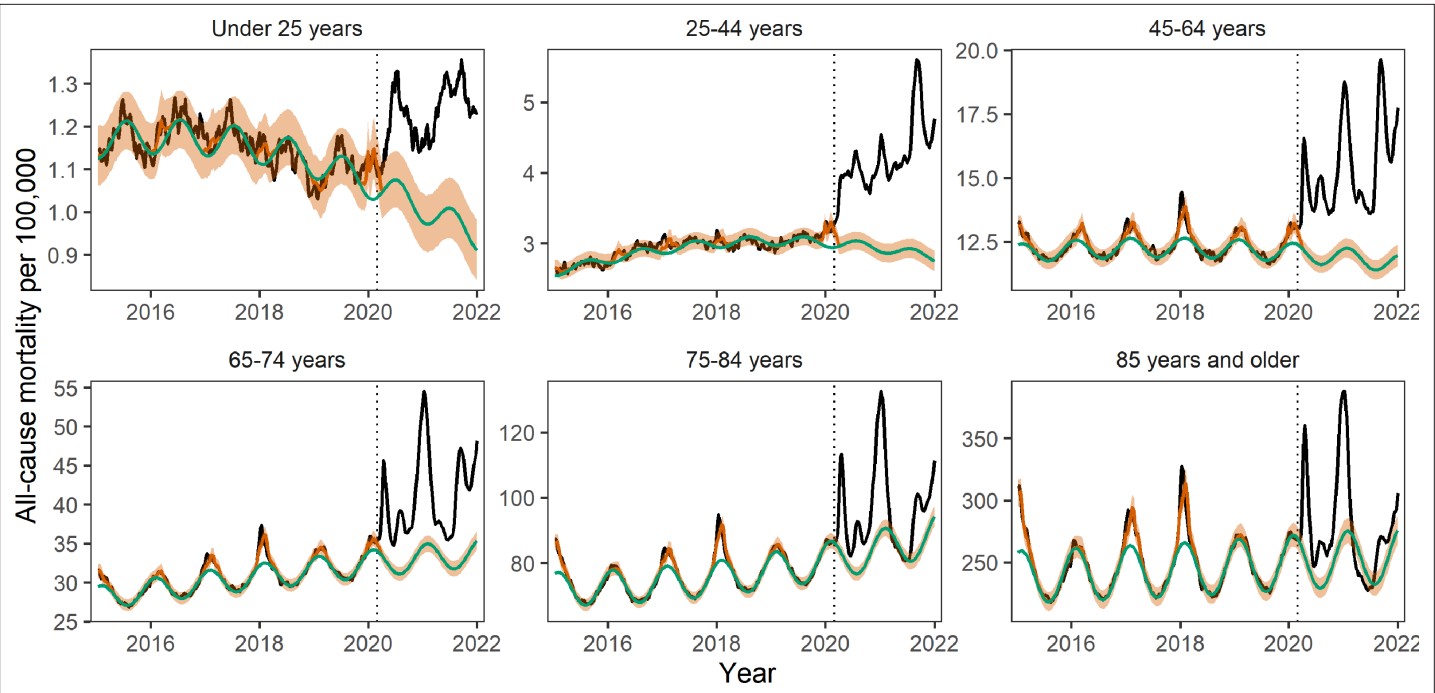

**Figure 5.** Age-specific weekly all-cause mortality rates time series per 100,000. The black line shows observed data, the green line shows the seasonal model baseline, the orange shaded areasthe 95% Confidence Interval (CI) on the seasonal baseline, and the red line shows model predictions with seasonal variation and influenza circulation. Excess mortality attributed to the COVID-19 pandemic is defined as the area between the black and green line from March 1, 2020 onwards. The dotted black vertical line marks the start of the pandemic on March 1, 2020.

## Pandemic age mortality patterns

Next, we ran some of the same analyses on age-specific data. The total burden and direct impacts of the COVID-19 pandemic from March 1, 2020 to January 1, 2022 varied substantially by age (*Figure 5*, *Table 4*). As many prior studies have reported, all-cause excess death rates increased monotonically with age. Individuals 85 years and older, the age group with the highest death rate, accounted for 17% of excess mortality, while individuals under 25 years accounted for only 2.1%. Age groups 45–64 years

**Table 4.** Age specific mortality patterns from March 1, 2020 to January 1, 2021: excess all-cause deaths, official COVID-19 deaths, and direct contribution of COVID-19 to mortality estimated by statistical model.

| Age | No. estimated excess all-cause deaths (95% prediction interval) | Estimated excess all-cause deaths per 100,000 (95% prediction interval) | Official statistics on the no. reported COVID-19 deaths* | Percent of excess deaths coded as COVID-19 in official statistics (%)† | Model estimate of % of excess deaths directly attributed to COVID-19‡ |
|---|---|---|---|---|---|
| *Under 25* years | 22,400 (15,400–29,000) | 21.4 (14.7–27.7) | 2853 | 0.13 (0.1–0.19) | −0.45 (−0.63, 0.07) |
| *25–44* years | 112,200 (100,200–123,100) | 125 (112–138) | 34,048 | 0.30 (0.28–0.34) | 0.02 (−0.09, 0.11) |
| *45–64* years | 286,500 (255,100–315,600) | 342 (304–376) | 183,284 | 0.64 (0.58–0.72) | 0.43 (0.35, 0.49) |
| *65–74* years | 254,900 (222,600–285,400) | 771 (673–863) | 194,436 | 0.76 (0.68–0.87) | 0.67 (0.59, 0.76) |
| *75–84* years | 189,200 (149,400–228,400) | 1137 (898–1373) | 217,479 | 1.15 (0.95–1.46) | 1.02 (0.88, 1.32) |
| *85 years and over* | 182,200 (120,400–242,500) | 2708 (1789–3605) | 218,464 | 1.20 (0.9–1.82) | 1.20 (1.02, 1.41) |

*Death certificates have multiple causes of death listed; here COVID-19 can be listed anywhere on the death certificate. These are deaths reported between March 1, 2020 to January 1, 2022, as available on June 14, 2022.

†Estimated as the proportion of excess all-cause deaths captured by official COVID-19 statistics (column 4 divided by column 2).

‡Proportion of all-cause excess mortality that is attributable to SARS-CoV-2 infection (direct pandemic impact) based on GAM models, where all-cause excess mortality is regressed against COVID-19 intensity, strength of interventions, and ICU occupancy each week. Estimates are based on comparison of predictions from the full model with counterfactual predictions where the COVID-19 term is set to zero.

and 65–75 years each accounted for a quarter of all-cause excess deaths estimated for the pandemic period.

Comparison of official COVID-19 statistics with excess mortality data provides empirical estimates of the direct impacts of the pandemic. The ratio of official COVID-19 statistics to all-cause excess deaths increased gradually with age, with official COVID-19 deaths representing only 13% (95% CI 10–19%) of excess mortality in individuals under 25 years and over 100% in the two oldest age groups (*Table 3*). The reported age gradient is consistent with a larger direct effect of the pandemic in older age groups; further, official statistics identified more deaths in seniors than estimated by excess mortality models.

To further investigate age differences in the direct and indirect effects of the pandemic, we applied our GAM approach to age-specific data (*Appendix 1—figures 18 and 19*). The COVID-19 term measuring the direct impact of SARS-CoV-2 infection on mortality was statistically significant in all age groups above 45 years but not in younger age groups. In contrast, the relative contribution of indirect effects, via the intervention variable, was highest in the youngest age groups and decreased with age. Overall, the direct contribution of COVID-19 to excess mortality was estimated to increase with age, from negative and non-statistically significant in individuals under 25 years to over 100% in those over 85 years, echoing the gradient seen in official statistics (*Table 4*). It is also worth noting that our excess mortality estimates may be too conservative (too high) as we did not account for missed circulation of endemic pathogens (see methods and Appendix). This could explain why our estimates of direct COVID-19 contribution exceed 100% in the oldest age groups. Furthermore, periods of stricter interventions were statistically associated with increased mortality in younger individuals, independently from the effect of SARS-CoV-2 infection.

Finally, to better understand the interplay between indirect mortality in younger age groups and deaths from external causes, we visualized age-specific monthly statistics on external deaths. These data were available for a subset of the study period (see *Appendix 1—figure 20*). The rise in mortality from external causes was concentrated in ages 15–44 years, with a notable elevation in May to July 2020 compared to 2019 levels. By mid-2021, mortality had substantially decreased, although some of the decreases could be attributable to reporting delays.

## Discussion

In this US study, we aimed to disentangle the direct and indirect mortality impacts of the COVID-19 pandemic from March 1, 2020 to January 1, 2022 using regression models and synchronicity analyses. We find that 84% (65–94%) of the rise in all-cause mortality during this period can be statistically linked to SARS-CoV-2 activity, lending support to the predominance of the direct mortality consequences of the pandemic on a national scale. We also find a direct contribution of SARS-CoV-2 infection to mortality from several chronic conditions such as Alzheimer's, diabetes, and heart diseases. This contribution is not captured in official statistics that consider COVID-19 as the primary cause of death. In contrast, analysis of mortality in children and young adults, and mortality from accidents and injuries, drug overdoses, assaults, and homicides, paints a different picture. Modeling of these death strata indicates a marked relationship with proxies for the strength of interventions, supporting a dominant contribution of indirect pandemic effects unrelated to SARS-CoV-2 infection. In contrast to other causes of death studied, cancer and suicides remained within baseline levels during the pandemic period.

Perhaps the most striking finding of our study is the large mortality burden of the pandemic in individuals 25–44 years, with an estimated 112,200 (100,200–123,100) excess deaths by January 1, 2022. Only 30% of these excess deaths are ascribed to COVID-19 in official statistics. Accordingly, our regression analysis does not support a predominant contribution of SARS-CoV-2 infection in this age group. The trajectory of mortality in this age group is disjoint from periods of intense COVID-19 circulation and statistically tied to a variable monitoring the strength of interventions. This finding supports a possible detrimental effect of COVID-19 control measures beyond the initial lockdown period in Spring 2020, although this is an ecological study that cannot prove causality nor elucidate the mechanisms at play. And while individuals under 25 years had a low overall excess death rate during the pandemic, we find that the contribution of indirect pandemic effects is even greater in this age group. In contrast, individuals over 65 years predominantly suffered from the direct consequences of SARS-CoV-2 infection. In a study of excess mortality in over 100 countries, *Karlinsky and Kobak, 2021* note

a predominance of the direct mortality consequences of the pandemic; however, they did not study age patterns. Our analysis shows that the contributions of direct and indirect effects vary with age and can skew towards indirect effects in the young, even in countries that experienced relatively high infections rates like the United States.

Prior studies have shown that a decrease in emergency visits for diabetes, stroke, and myocardial infarctions in all age groups coincided with a rise in mortality for these conditions (*Lange et al., 2020*). Faust et al., estimated that 38% of deaths between 25 and 44 years were due to COVID-19 during March to July 2020, compared to 26% (23–30%) in our study that considers a much longer time period (*Faust et al., 2021a*). Public health interventions, limited medical care, and behavioral changes (e.g. delays in seeking timely medical help due to fear of infection, *Bollmann et al., 2020*; *Kansagra et al., 2020*; *Mafham et al., 2020*; *Woolf et al., 2021*) could have contributed to the surge in excess deaths unrelated to COVID-19 in young adults, resulting in a notable peak of mortality in summer 2020. In addition, we find that mortality from external causes remained elevated during May to August 2020 among young adults, likely driven by an elevation in deaths from opioid poisonings, accidents, and assaults.

Mortality from external causes increased by 102,800 (81,400–123,700) from March 1, 2020 to January 1, 2022. There was a moderate correlation with pre-pandemic baseline mortality rates in state-level data (*Appendix 1—figure 13*), indicating that states with historically high death rates from external causes experienced more prominent increases during the pandemic. The rise in external cause mortality was most pronounced in the subcategory of assaults and homicide, followed by overdoses and accidents. In contrast, mortality from suicides remained stable or below expectations. Prior work by *Faust et al., 2021b*, reported a decrease in suicide in several countries during March to July 2020, including in the United States (10%); here, we show that suicide mortality remained stable throughout the rest of 2020–2021. Furthermore, *Faust et al., 2021a* reported that deaths from overdoses and injuries increased during March-July 2020. In our data, the increase persisted until the end of 2021. Overall, of the eight causes of death studied here, the indirect pandemic effects were statistically largest in external mortality causes.

There was strong synchronicity between respiratory mortality and mortality from other conditions during spring 2020, possibly due to poor SARS-CoV-2 test availability and guidelines to restrict testing to just hospitalized cases in the early pandemic stages. Many deaths in nursing homes or at home during March to April 2020 were never tested, and they were recorded as known unlying conditions (i.e. heart disease, Alzheimer's, and diabetes) by default. In addition, Alzheimer patients typically live in long-term care facilities and may have been at increased risk of (untested) COVID-19 infection early in the pandemic. Interestingly, the correlation between excess mortality from respiratory diseases and Alzheimer increased in the winter 2020–2021 wave, signaling a persistent direct impact of COVID-19 on Alzheimer's in a period where COVID-19 incidence and testing propensity were high.

We validated our excess mortality estimates against serology and assessed the IFR, a parameter notoriously difficult to measure. Our all-age estimate of 0.67% (95% CI 0.60–0.73%) is consistent with a 2020 meta-analysis (*Meyerowitz-Katz and Merone, 2020*) and an early study from China (0.66%; 95% CI: 0.39—1.33%) (*Verity et al., 2020*). A study of all-cause excess mortality in the Netherlands reports a substantially higher IFR (1%) (*van Asten et al., 2021*); however, all-cause mortality is not specific to COVID-19 (accordingly, our estimate based on all-cause mortality is higher at 0.89% (95% CI 0.77–1.02%)). IFR estimates based on official COVID-19 statistics were 15% higher than those based on excess respiratory mortality, yet the official statistics did not correlate with serology data as well as with our excess mortality estimates. Between-state differences in COVID-19 death coding practices could explain these findings. Overall, our analyses support the robustness and specificity of excess respiratory mortality (with the addition of the COVID-19 specific code) as an indicator of the COVID-19 mortality burden.

A few states were outliers in the regression of excess mortality against serology; for instance, New York had a higher than predicted IFR, while Michigan had a lower than predicted IFR. Several non-mutually exclusive factors could drive these findings, including a higher proportion of deaths among older individuals (aligned with the demography of New York state), large outbreaks in long-term care facilities, lack of knowledge on the management of severe patients early in the pandemic (*Barnett et al., 2020*), and waning immunity. Serosurveys conducted in December 2021 could underestimate cumulative SARS-CoV-2 infection rates in states that have experienced most of their infections in early

2020. All states contributing to CDC serology surveillance used the Roche assay to test for the presence of SARS-CoV-2 antibodies, which is less prone to waning than other assays but has some decay (*Prete et al., 2022*). We ran a sensitivity analysis using the maximum seroprevalence recorded over the study period rather than the seroprevalence at the end of the study period (*Appendix 1—figure 10*); New York remained an outlier in this analysis.

The roll-out of a large SARS-CoV-2 vaccination campaign starting in December 2020 in the United States has had a major impact on rates of hospitalizations and deaths for COVID-19. Yet, excess mortality from all causes and respiratory diseases has remained elevated all throughout 2021, fueled by new SARS-CoV-2 variants, particularly Delta and Omicron. COVID-19 mortality is now concentrated among unvaccinated groups, with the highest vaccination rates reported in older individuals (*Centers for Disease Control and Prevention, 2022h*). Coincidentally, in our data, excess respiratory mortality started to decouple from chronic condition excess mortality (e.g. diabetes, Alzheimer, heart diseases) around the time of the Delta wave in the summer of 2021. Similarly, among the oldest and most highly vaccinated age groups, the Delta wave had a proportionally milder impact than the wave dominated by the ancestral strain in winter 2020–21, while the pattern was reversed in younger age groups. Decoupling between COVID-19 mortality and mortality from chronic conditions is likely due to vaccination rather than a specific variant. Yet decoupling remains insufficient to obfuscate synchronicity in these conditions observed during the full study period from March 1, 2020 to January 1, 2022. Further analyses of the Omicron period could reveal different synchronicity patterns.

It is interesting that during April-June 2021, and before the rise of the Delta variant, mortality from cancer, Alzheimer and heart diseases was below baseline (negative excess mortality), with a similar phenomenon observed for all-cause mortality among individuals 75–84 and over 85 years. These negative excesses could signal a displacement of the mortality baseline, whereby frail individuals are harvested by a large-scale infectious disease event, resulting in a decline in baseline mortality in the aftermath -- similar patterns have been associated with heatwaves (*Saha et al., 2014*). Harvesting is also consistent with our regression analysis, where estimates of the direct impacts of COVID-19 exceeded 100% in the two oldest age groups (albeit with broad confidence intervals, *Table 4*). This would be expected if the baseline mortality was overestimated due to harvesting. Furthermore, the age profile of COVID-19 severity risk dictates that older individuals would bear the strongest effects of harvesting, which is consistent with the US data. A competing hypothesis for an inflated baseline is the role of non-COVID-19 pathogens that were repressed during the pandemic due to social distancing. These pathogens are implicitly included in our baseline model calibrated to pre-pandemic years. Although the timing of negative excesses (Spring 2021) is not fully supportive of the contribution of missed pathogens which tend to predominate in winter, harvesting and depressed pathogen circulation are two mechanisms inherently difficult to separate. A third competing mechanism is that official COVID-19 deaths capture deaths with COVID-19, rather than deaths from COVID-19. Although this is likely true to some extent, our estimates of direct effects exceed 100% in regression models that ignore official statistics, suggesting that other mechanisms are at play.

Our study is subject to several limitations. First, mortality counts below the minimum cut-off value of 10 were suppressed due to privacy regulations. As a result, our age-specific analyses were restricted to larger states, and we could not assess the role of race and ethnicity. Prior work has shown important disparities in COVID-19 impact by race/ethnicity and economic status in the United States and abroad (*Mena et al., 2021*; *Rossen et al., 2021*). Second, official coding practices may have changed between states and through time-based on SARS-CoV-2 testing availability, location of death, demographic factors, and comorbidities. Third, we assumed full coverage of death reporting, which may not be valid throughout the United States (*Murray et al., 2010*), and we did not study changes in deaths ascribed to ill-defined codes (R codes). Ill-defined deaths would be captured in all-cause mortality but not in cause-specific analyses. Fourth, we find periods of negative excesses in cancer (throughout the pandemic), cardiovascular, and heart diseases, possibly due to changes in the ascertainment of the underlying cause of death (e.g. death in a cancer patient with COVID-19 is ascribed to COVID-19), harvesting (*Saha et al., 2014*), or depressed circulation of endemic pathogens other than influenza. Fifth, we choose to fit the model baseline to data for 2014–2020, which is arbitrary. We studied the sensitivity of our excess mortality estimates to this assumption (*Appendix 1 - Figure 9*). While national analyses were robust to this choice, as well as state-level analyses of most conditions, state-specific estimates for Alzheimer's disease were more sensitive. Furthermore, the

model baseline did not account for possible point-in-time disasters that may have occurred during the pandemic but are independent of COVID-19 (e.g. a hurricane) or changes in air pollution. Sixth, GAM indicate that the Oxford stringency index, used as a proxy for the strength of interventions, is a dominant predictor of excess mortality from external causes and all-cause mortality in individuals under 45 years. Yet, the relationships are non-linear, and the resulting models do not fully capture mortality changes during the pandemic. Along the same lines, the Oxford stringency index does not consider the actual implementation nor the effect of interventions; it is solely based on mandates in place in different locations and time periods. We also assume that, for a given level of stringency, the impact of interventions does not change over time. Because time and intervention stringency are highly conflated, it would be difficult to study potential temporal variation in this relationship. Furthermore, analyses are aggregated at the state or national level, while implementation of interventions may operate more locally. We also do not account for underlying differences in vulnerability between states, where more vulnerable states may have implemented stricter interventions (although this potential bias would not affect temporal analyses). Finally, our study ends on January 1, 2022 and does not capture a recrudescence of COVID-19-related deaths due to the Omicron variant. As a result, our excess mortality estimates should be deemed conservative.

Pandemic excess mortality patterns have been heterogeneous globally (*Islam et al., 2021*; *Karlinsky and Kobak, 2021*; *Kontis et al., 2020*; *Nørgaard et al., 2021*). In a comprehensive analysis of mortality in 21 countries in Europe, New Zealand, and Australia, official COVID-19 deaths accounted for an average of 77% (62–93%) of all-cause excess deaths during the first wave of the pandemic from March to May 2020 (*Kontis et al., 2020*). However, there was a greater disconnect between estimates in hard-hit countries such as the UK, Spain, Italy, and Belgium (*Kontis et al., 2020*; *Kontopantelis et al., 2021*; *Odone et al., 2021*). In a study of over 100 countries, the ratio between excess mortality and official deaths was 1.6 on average, but went as high as 50 (*Karlinsky and Kobak, 2021*). The disconnect was primarily attributed to the under-detection of COVID-19 rather than indirect effects, although indirect effects were not explicitly modeled. In Italy, the case fatality rate for acute myocardial infarction increased threefold during the first wave, while hospitalizations for these conditions decreased by 48% (*Odone et al., 2021*), suggesting that at least some of the excess deaths were indirect deaths. In the UK, cancer deaths increased about 10% at the height of the April 2020 lockdown; however, more recent fluctuations in cancer mortality remain unclear (*Lai et al., 2020*). Interestingly, in New Zealand, where control of COVID-19 has been remarkable, mortality was slightly but not significantly below baseline (*Kontis et al., 2020*). A similar finding was described in Russian provinces where a lockdown was implemented before the onset of COVID-19 (*Kobak, 2021*). This suggests that a lockdown without COVID-19 is neither preventing nor causing an appreciable number of deaths, although effects could be country-dependent. In the United States, the direct impacts of the pandemic greatly outweigh the indirect consequences in all age data, but the reverse is true in children and young adults. Further work should concentrate on comparing the direct and indirect impact of COVID-19 in different countries over the same time period and using the same methodology.

## Conclusion

Here, we examined trends in cause-specific mortality across states and age groups to address the direct and indirect impacts of COVID-19 in the United States. We find that 84% (95% CI 65–94%) of the total mortality elevation during the March 1, 2020 to January 1, 2021 pandemic period is attributable to the direct impact of SARS-CoV-2 infection. There is, however, a large indirect impact of the pandemic on children and young adults, and on mortality from external causes, particularly from accidents, assaults, and overdoses and these indirect impacts are statistically linked to indicators of the strength of interventions. We also find an undetected contribution of SARS-CoV-2 infection on mortality from chronic conditions, such as Alzheimer's, diabetes, and heart diseases, which has not fully disappeared after 2 years of SARS-CoV-2 circulation and a large vaccination program. Our conclusions are based on ecologic analyses that are useful for generating hypotheses but do not prove causality. As more detailed information becomes available with the release of individual death certificates, it will be important to dissect the drivers of mortality among younger adults and certain ethnic groups, and understand how chronic conditions, violence, opioids, and suicides intersect with large-scale infectious disease events and behavioral changes.

## Materials and methods

### Mortality data

We obtained weekly mortality counts from the National Center for Health Statistics (NCHS) for the period August 1, 2014 to January 1, 2022 (the last week of 2021 ended on January 1st, 2022); we included 2014–2019 data to construct robust historical model baselines (*Centers for Disease Control and Prevention, 2022a*; *Centers for Disease Control and Prevention, 2022b*). Data were stratified by state, six age groups (all ages, under 25 years, 25–44, 45–64, 65–74, 75–84, and over 85), and eight underlying mortality causes (all causes, respiratory conditions, Alzheimer's disease, cancer, cerebrovascular diseases, diabetes, heart disease, external causes; see supplement for disease codes). External causes include suicides, accidents, homicides, and poisoning from opioids and other substances, among other conditions. We used aggregated mortality counts ascribed to COVID-19 (ICD code U07), influenza, pneumonia, and chronic lower respiratory diseases as an indicator of 'respiratory mortality,' which was our most specific indicator of excess deaths directly attributable to SARS-CoV-2 infection. Furthermore, we compiled weekly deaths with any mention of COVID-19 anywhere in the death certificate and considered those to be the official COVID-19 statistics (*Centers for Disease Control and Prevention, 2022c*).

Weekly mortality is available with a typical lag of 7 weeks. We undertook this work more than 7 months after the last observation, ensuring that there was little reporting delay for the data presented in this study. As a result, we did not apply any backfilling algorithms, unlike prior work (*Weinberger et al., 2020*).

To further explore patterns in external mortality causes, which include a range of conditions, we obtained additional monthly data by subcategories of deaths, including suicides, assaults and homicides, drug overdoses, accidents and unintentional injuries, and motor vehicle accidents (a subset of accidents) (*Centers for Disease Control and Prevention, 2022d*). We also downloaded monthly deaths from external causes combined by age and region (*Centers for Disease Control and Prevention, 2022e*). External causes of death are typically released several months later than other conditions; detailed data are unavailable at a weekly resolution.

### Other datasets

Age- and state-specific population estimates were obtained from CDC (*Centers for Disease Control and Prevention, 2022f*) and used to calculate mortality rates. To validate our mortality approach, we compared our excess mortality estimates with serology, using the CDC's state-specific observations from the 28th round of SARS-CoV-2 serology surveys (*Centers for Disease Control and Prevention, 2022g*). These surveys provide estimates of the proportion of the population with SARS-CoV-2 antibodies to the nucleocapsid by late December 2021, which is a measure of cumulative infections. As the nucleocapsid antigen is not a component of the vaccines used in the United States, the serologic assay only captures natural infections. We compared these serologic estimates with our estimates of cumulative excess death rates on January 1, 2022, assuming a similar delay between infection and death, and between infection and antibody rise. This comparison was also used to estimate the IFR. We ran a sensitivity analysis on the maximum seroprevalence reported during the study period, rather than seroprevalence at the end of the study period, to account for the potential waning of natural immunity.

To adjust excess mortality models for the contribution of influenza, we used weekly data on influenza circulation based on CDC surveillance (*Rudis et al., 2021*). To evaluate the putative impacts of public health interventions on cause-specific mortality, we compiled the health containment index from the COVID-19 government response tracker, which measures the strength of interventions by week and state (*Oxford University, 2021*). Furthermore, to evaluate how hospital strain may have contributed to excess mortality, we used the HHS COVID-19 dataset on hospital use, which has hospital-level indicators of ICU bed utilization (*Health and Human Services, 2022*). Data were aggregated by week and state.

All data used in the analysis were publicly available and exempt from human subject review; the data and code have been posted in a GitHub repository (https://github.com/viboudc/DirectIndirectCOVID19MortalityEstimation).

### Analytic approach

## Excess mortality models

To estimate seasonal mortality baselines, we applied linear regression models to weekly mortality rates in the pre-pandemic period, August 1, 2014 to March 1, 2020; excess mortality was the difference between observed mortality and baseline mortality (*Goldstein et al., 2012*; *Weinberger et al., 2020*; see Appendix for more details). Baselines were estimated separately for each mortality cause, age group, and state. The models included harmonic terms for seasonality, time trends, and a proxy for weekly influenza incidence. In this approach, we single out influenza by explicitly modeling the contribution of this virus on mortality and taking influenza mortality out of the mortality baseline. Hence, our estimates of excess deaths due to COVID-19 are deaths above the baseline from prior years, after the impact of influenza has been removed. Our choice is motivated by the fact that influenza can cause large mortality variations between years due to differences in circulating strains; hence, it is not straightforward to define an average influenza season as part of the baseline. In contrast, we do not explicitly model the mortality contribution of other pathogens, for which we do not have surveillance data, and which we assume to be less variable between years. The mortality contribution of these pathogens implicitly becomes part of the baseline. Since, the circulation of endemic pathogens has been greatly reduced during the pandemic, our baseline is likely inflated, leading to conservative estimates of excess mortality. We return to this question in the discussion.

We fitted the model to data until March 1, 2020 and projected the baseline forward until January 1, 2022. We estimated weekly excess mortality by subtracting the predicted baseline from the observed mortality that week; total excess mortality was the sum of weekly excesses (positive or negative) from March 1, 2020 to January 1, 2022. We used block bootstraps to generate 95% uncertainty intervals on excess mortality estimates.

We ran cause-specific excess mortality analyses nationally and for states that had sufficient mortality counts, as weekly death counts below 10 were suppressed due to privacy concerns. States missing more than 2 weeks of data between March 1, 2020 and January 1, 2022 were excluded from the corresponding analyses. We ran respiratory excess mortality analyses for 16 states and non-respiratory analyses for 33 states (see Appendix for full list).

For reference, we compared the mortality impact of COVID-19 with that of a severe influenza season. This comparison is not straightforward as an influenza epidemic typically lasts 3–4 months while COVID-19 mortality has persisted for over 2 years. Yet, to provide context, we compared excess mortality during November to March of 2017–2018, corresponding to a recent severe influenza season dominated by the A/H3N2 virus, with the same months in 2020–2021, which correspond to the largest national wave of COVID-19.

## Estimation of direct and indirect pandemic impacts

To assess the direct and indirect consequences of the pandemic on mortality, we performed several correlation and regression analyses evaluating the trajectory of different causes of deaths by age and geography, building on earlier work on the 1968 influenza pandemic (*Reichert et al., 2004*) and COVID-19 (*Sharma et al., 2021*) (see Appendix for details). First, we tested whether weekly respiratory excess mortality became increasingly correlated with other causes of death during the pandemic period March 1, 2020 to January 1, 2022, compared to pre-pandemic periods of similar duration. An increase in correlation would signal a direct but undetected effect of COVID-19 on non-respiratory mortality.

Second, we assessed whether states that experienced high cumulative COVID-19 mortality experienced high cumulative mortality from other causes during the pandemic. We used our estimates of excess respiratory mortality and official COVID-19 death tallies, as complementary measures of COVID-19 mortality.

Third, we estimated the fraction of excess mortality attributable to the direct impact of SARS-CoV-2 infection, vs the indirect impacts that are driven by non-pharmaceutical interventions and hospital strain. We regressed weekly excess mortality against COVID-19 deaths, the health containment index, and ICU use, after exploring different lags between predictors and outcomes. We used GAM to allow for non-linear effects between mortality and all covariates. Models were run separately for each cause of death, nationally and by state, and for each age group. Uncertainty in weekly excess mortality estimates was propagated into the regression models. Attributable fractions were

obtained by resampling from the models after setting covariates to their minimum values reported for the pandemic period; for instance, the attributable fraction for COVID-19 was based on the relative difference in cumulative predicted mortality when the COVID-19 variable was at its observed values vs. zero, with the other covariates remaining unchanged (Appendix).

## Validation of excess deaths based on serology and estimation of IFR

Since COVID-19 deaths are ultimately the result of SARS-CoV-2 infections, serology can validate the accuracy of excess mortality estimates. To test the validity of the excess mortality approach for our most proximal mortality indicator of COVID-19, we regressed cumulative excess respiratory mortality rates against SARS-CoV-2 seroprevalence estimates at the state level. We used a model without intercept since we assumed a direct correspondence between rates of infection and death. We repeated this analysis with all-cause excess mortality, official COVID-19 deaths, and excess mortality in individuals over 65 years as the outcome variable. We used this analysis to estimate the IFR, based on the slope of the above regressions. We propagated the errors obtained in excess mortality and seroprevalence estimates into IFR estimates (Appendix).

## Acknowledgements

DMW acknowledges funding from NIH/NIAID (R01AI137093). LS acknowledges funding from the Carlsberg foundation.

## Additional information

### Funding

| Funder | Grant reference number | Author |
|---|---|---|
| National Institute of Allergy and Infectious Diseases | R01AI137093 | Daniel M Weinberger |
| Carlsbergfondet | | Lone Simonsen |

The funders had no role in study design, data collection and interpretation, or the decision to submit the work for publication.

### Author contributions

Wha-Eum Lee, Data curation, Formal analysis, Visualization, Methodology, Writing – original draft, Writing – review and editing; Sang Woo Park, Supervision, Methodology, Writing – review and editing; Daniel M Weinberger, Conceptualization, Software, Methodology, Writing – review and editing; Donald Olson, Conceptualization, Methodology, Writing – review and editing; Lone Simonsen, Conceptualization, Writing – review and editing; Bryan T Grenfell, Conceptualization, Supervision, Writing – review and editing; Cécile Viboud, Conceptualization, Data curation, Formal analysis, Visualization, Methodology, Writing – original draft, Writing – review and editing

### Author ORCIDs

Sang Woo Park (ID) http://orcid.org/0000-0003-2202-3361
Daniel M Weinberger (ID) http://orcid.org/0000-0003-1178-8086
Lone Simonsen (ID) http://orcid.org/0000-0003-1535-8526
Bryan T Grenfell (ID) http://orcid.org/0000-0003-3227-5909
Cécile Viboud (ID) http://orcid.org/0000-0003-3243-4711

### Decision letter and Author response

Decision letter https://doi.org/10.7554/eLife.77562.sa1
Author response https://doi.org/10.7554/eLife.77562.sa2

## Additional files

### Supplementary files
• Transparent reporting form

### Data availability

Our study presents an analysis of publicly available surveillance and mortality data, so no new data have been generated for this manuscript. The modelling code and underlying data have been posted in the following public Github repository https://github.com/viboudc/DirectIndirectCOVID19MortalityEstimation (copy archived at swh:1:rev:f0c23c87a00f589179af5351e89732f51d5a2b19).

The following previously published datasets were used:

| Author(s) | Year | Dataset title | Dataset URL | Database and Identifier |
|---|---|---|---|---|
| Centers for Disease Control and Prevention, National Center for Health Statistics | 2022 | Weekly Counts of Deaths by State and Select Causes, 2014-2019 | https://data.cdc.gov/NCHS/Weekly-Counts-of-Deaths-by-State-and-Select-Causes/3yf8-kanr | National Center for Health Statistics, Weekly-Counts-of-Deaths-by-State-and-Select-Causes/3yf8-kanr |
| Centers for Disease Control and Prevention, National Center for Health Statistics | 2022 | Weekly Counts of Deaths by State and Select Causes, 2020-2023 | https://data.cdc.gov/NCHS/Weekly-Counts-of-Deaths-by-State-and-Select-Causes/muzy-jte6 | National Center for Health Statistics, Weekly-Counts-of-Deaths-by-State-and-Select-Causes/muzy-jte6 |
| Centers for Disease Control and Prevention, National Center for Health Statistics | 2022 | Provisional COVID-19 Deaths by Week, Sex, and Age | https://data.cdc.gov/NCHS/Provisional-COVID-19-Deaths-by-Week-Sex-and-Age/vsak-wrfu | National Center for Health Statistics, Provisional-COVID-19-Deaths-by-Week-Sex-and-Age/vsak-wrfu |
| Centers for Disease Control and Prevention, National Center for Health Statistics | 2022 | Monthly Counts of Deaths by Select Causes, 2014-2019 | https://data.cdc.gov/NCHS/Monthly-Counts-of-Deaths-by-Select-Causes-2014-201/bxq8-mugm | National Center for Health Statistics, Monthly-Counts-of-Deaths-by-Select-Causes-2014-201/bxq8-mugm |
| Centers for Disease Control and Prevention, National Center for Health Statistics | 2022 | Monthly Provisional Counts of Deaths by Select Causes, 2020-2022 | https://data.cdc.gov/NCHS/Monthly-Provisional-Counts-of-Deaths-by-Select-Cau/9dzk-mvmi | National Center for Health Statistics, Monthly-Provisional-Counts-of-Deaths-by-Select-Cau/9dzk-mvmi |
| Centers for Disease Control and Prevention, National Center for Health Statistics | 2022 | AH Monthly Provisional Counts of Deaths by Age Group and HHS region for Select Causes of Death, 2019-2022 | https://data.cdc.gov/NCHS/AH-Monthly-Provisional-Counts-of-Deaths-by-Age-Gro/ezfr-g6hf | National Center for Health Statistics, AH-Monthly-Provisional-Counts-of-Deaths-by-Age-Gro/ezfr-g6hf |
| Centers for Disease Control and Prevention, National Center for Health Statistics | 2022 | State Population Projections 2004-2030 | https://wonder.cdc.gov/population-projections.html | Centers for Disease Control, WONDER |
| Centers for Disease Control | 2022 | Nationwide COVID-19 Infection-Induced Antibody Seroprevalence (Commercial laboratories) | https://covid.cdc.gov/covid-data-tracker/#national-lab | Centers for Disease Control, covid-data-tracker/#national-lab |
| Centers for Disease Control | 2022 | COVID data tracker | https://covid.cdc.gov/covid-data-tracker/#datatracker-home | Centers for Disease Control, covid-data-tracker/#datatracker-home |

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

## Appendix 1

### Supplemental data and methods
General approach

Our analysis relies on modeling of weekly trends in US death certificates compiled from the National Center for Health Statistics (NCHS) website (*Centers for Disease Control and Prevention, 2022a*; *Centers for Disease Control and Prevention, 2022b*; *Centers for Disease Control and Prevention, 2022c*). The goal of the study is to estimate the direct mortality impact of COVID-19, which results from SARS-CoV-2 infection, from the indirect impacts of the pandemic which can be linked to societal or health-related changes brought about by the pandemic.

We define direct COVID-19 mortality as the sum of deaths confirmed and coded as COVID-19 as the underlying cause, deaths coded as another underlying primary cause but with COVID-19 as one of the contributing multiple causes, and deaths that were directly caused by SARS-CoV-2 infection but for which a COVID-19 code is not included in the death certificate due to misdiagnosis or lack of testing. The first two categories are included in official death tallies of COVID-19 (since a COVID-19 code appears in the death certificate), while the third category can only be estimated using excess mortality approaches.

We define the indirect mortality impact of the pandemic as the sum of deaths due to healthcare avoidance or inaccessibility, deaths due to conditions or events that are exacerbated by non-pharmaceutical interventions and other pandemic behavior, and would not have occurred otherwise (i.e. suicide, drug overdose, homicide, or a stressed healthcare system that is unable to treat conditions unrelated to SARS-CoV-2).

It is worth noting that circulation of multiple pathogens has plummeted due to social distancing interventions in 2020–2021, and therefore the pandemic could have prevented a number of infectious disease deaths relative to historical expectations. Excess mortality approaches (*Weinberger et al., 2020*; *Goldstein et al., 2012*) will capture the net sum of these indirect impacts. These direct and indirect mortality pathways are not mutually exclusive and plausible over a wide range of conditions. For example, SARS-CoV-2 infection may trigger death in a patient with diabetes, which may be missed by testing, with no COVID-19 code listed on the death certificate. Concomitantly, a diabetic patient without a recent history of SARS-CoV-2 infection may turn away from the healthcare system at the height of the pandemic and die from lack of treatment. Our analysis attempts to separate these effects.

### Mortality data
Mortality conditions studied and states selected for further analysis

We used the international classification of disease version-10 to retrieve deaths for the period from August 1, 2014 to January 1, 2022 for the following 8 mortality outcomes: All-cause (deaths from any causes), Alzheimer's (G30), Cancer (C00-C97), Cerebrovascular diseases (I60-I69), Diabetes (E10 – E14), Heart disease (I00-I09, I11,I13,I20-I51), Respiratory Conditions (J09-J18, J40-J47, U071[the code for COVID-19]), External Cause (V01-Y89, U01-U03). Deaths with any of these codes as the underlying cause of deaths were selected for analysis.

16 states were selected for further analysis of respiratory mortality because they had sufficient counts on a weekly basis: Alabama, Arizona, California, Florida, Georgia, Illinois, Indiana, Michigan, Missouri, New Jersey, New York, Ohio, Pennsylvania, Tennessee, Texas, and Virginia. Weekly deaths count below 10 are blanked by NCHS for privacy reasons; states that had more than 2 weeks of blanked observations during the pandemic period were excluded. For states that were retained for respiratory mortality analyses, we interpolated missing data (≤2 weeks of interpolated data over 388 study weeks). Respiratory mortality was our most restrictive mortality outcome; it is based on aggregation of deaths from pneumonia and influenza with deaths from other respiratory conditions. Pneumonia and influenza can be uncommon on a weekly basis in less populous states, especially in summer, and hence blanked observations are not uncommon in the state-level dataset.

We applied a similar reasoning to the 7 other causes of deaths that are unrelated to respiratory conditions. Death counts were more numerous for these conditions than for respiratory mortality. If a state had more than 2 blanked weeks for one of the conditions, it was excluded for analysis of the other conditions. The following 33 states were included for analysis of non-respiratory mortality: Alabama, Arizona, Arkansas, California, Colorado, Connecticut, Florida, Georgia, Illinois, Indiana,

Iowa, Kansas, Kentucky, Louisiana, Maryland, Massachusetts, Michigan, Minnesota, Mississippi, Missouri, Nevada, New Jersey, New York, Ohio, Oklahoma, Oregon, Pennsylvania, South Carolina, Tennessee, Texas, Virginia, Washington, and Wisconsin.

## Analytical approach

### Weekly excess mortality model

We applied negative binomial seasonal regression models to weekly cause- and age-specific mortality, using an identity link, inspired by prior work on COVID-19 (*Weinberger et al., 2020*). Models included time trends, harmonic terms for seasonality, and terms for influenza circulation, following:

$$y_t = \beta_0 + \beta_1 t + \beta_2 t^2 + \beta_3 \sin\left(\frac{2\pi t}{52.17}\right) + \beta_4 \cos\left(\frac{2\pi t}{52.17}\right) + \sum_{i=1}^{6} \alpha_i x_{\{i,t\}} + e_t$$

where $t$ = time
$\beta_0$ = intercept
$\beta_1$ and $\beta_2$ = time trends coefficients
$\alpha_i$ = flu coefficients in season $i$
$x_{i,t}$ = influenza proxy at time $t$, season $i$
$e_t$ = error terms

The proxy for weekly influenza incidences was calculated by multiplying the weekly percentage of physician visits for influenza-like illness and weekly percentage of positive influenza tests (*Bollmann et al., 2020*), which were obtained via CDC's FluView portal using the cdcfluview package in R (package version 0.9.1). We let the influenza coefficient vary each season to reflect a different mix of circulating subtypes, associated with different severities. Influenza data were not available for New Jersey and Florida; instead, we used data from New York state and HHS Region 4, respectively. Some states discontinue laboratory surveillance for influenza virus circulation during the summer months, so we replaced the missing summer weeks with zero. Weekly influenza incidences after March 1, 2020 were set to zero, in line with the minimal circulation of influenza reported in this time period . Before fitting the model, we smooth the mortality data by running a 5 week moving average (this choice does not affect respiratory or all-cause mortality, but can be particularly useful to stabilize some of the state-specific time series for chronic conditions). We compute CI following prior work on COVID-19 excess mortality in the United States (*Weinberger et al., 2020*).

March 1, 2020 was set as the start of the period of putative pandemic-related excess mortality. We fitted the model to data from August 1, 2014 to March 1, 2020 and projected the baseline forward until January 1, 2022. We estimated weekly excess mortality related to the COVID-19 pandemic by subtracting the predicted baseline from the observed mortality that week. Total excess mortality for the pandemic period was defined as the sum of weekly excesses (positive or negative) during March 1 to January 1, 2022.

We also tested different link functions and error structures for the model and report the best fit to data here. Weekly observations and model baselines are displayed for the cause of death and administrative area (United States and each state) in *Appendix 1—figures 1–8*.

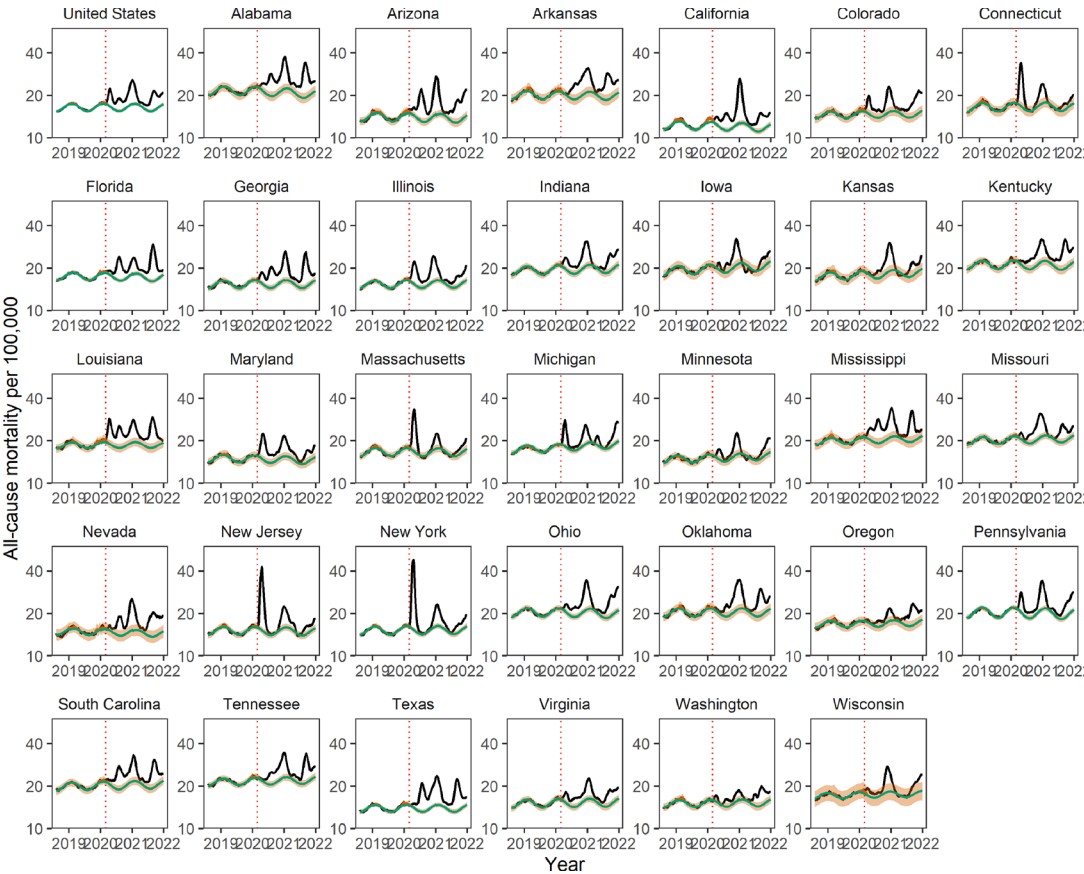

**Appendix 1—figure 1.** Trends in weekly all-cause mortality, nationally, and by state, August 1, 2018 to Janaury 1, 2022 Black lines show observed weekly death rates. Green lines show the seasonal baseline estimated based on time series regression of pre-pandemic data from August 1, 2014 to March 1, 2020 (the graph below truncates earlier years for sake of clarity). The red solid line shows the seasonal variation accounting for influenza circulation. The orange shading shows the upper and lower 95% confidence intervals (CIs) on the baseline. The dotted vertical line marks March 1, 2020. 33 states were selected for analysis of non-respiratory mortality causes based on completeness patterns.

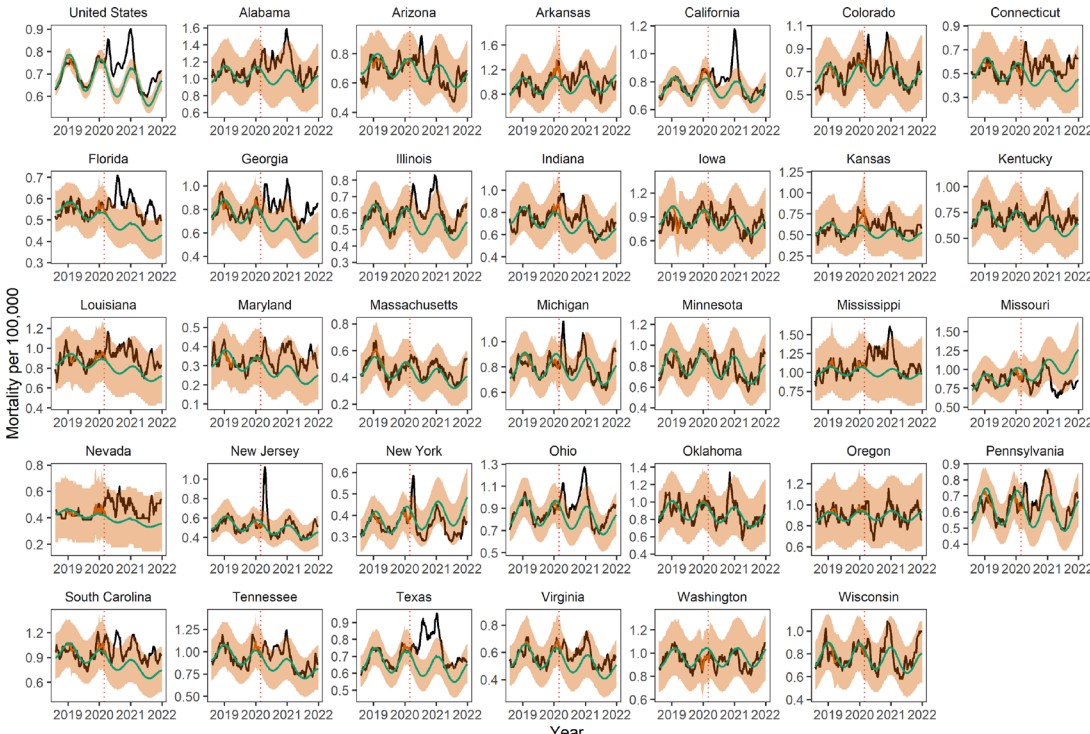

**Appendix 1—figure 2.** Trends in weekly Alzheimer's mortality, nationally, and by state. Legend as in *Appendix 1—figure 1*

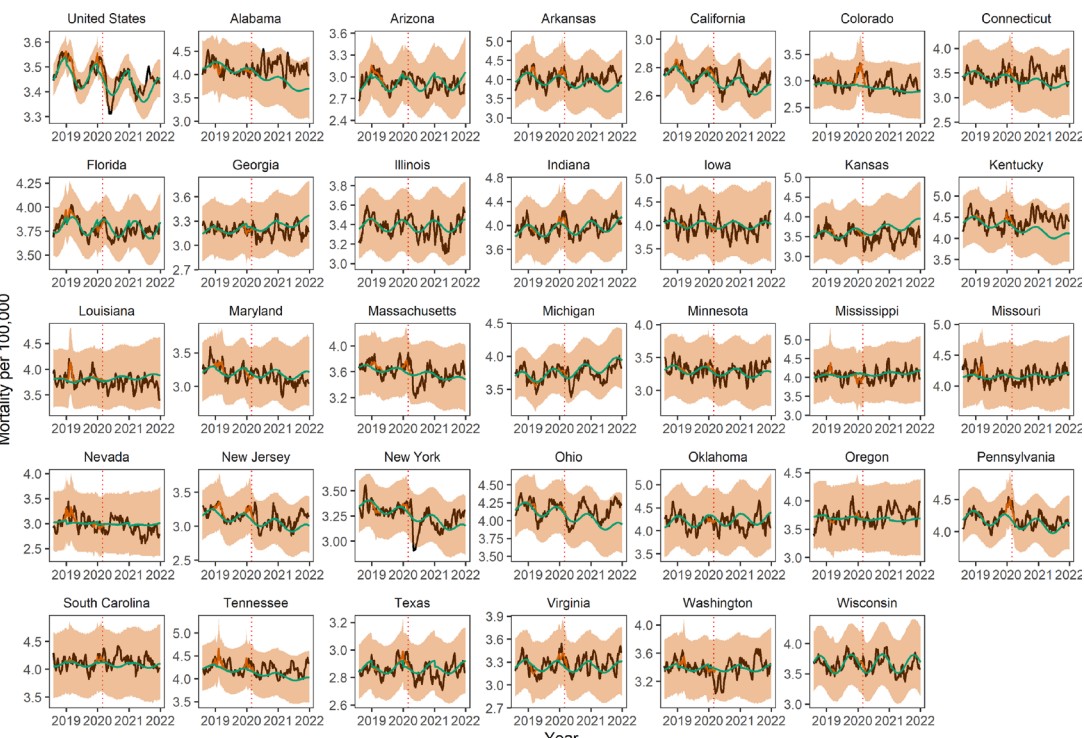

**Appendix 1—figure 3.** Trends in weekly cancer mortality, nationally, and by state. Legend as in *Appendix 1—figure 1*

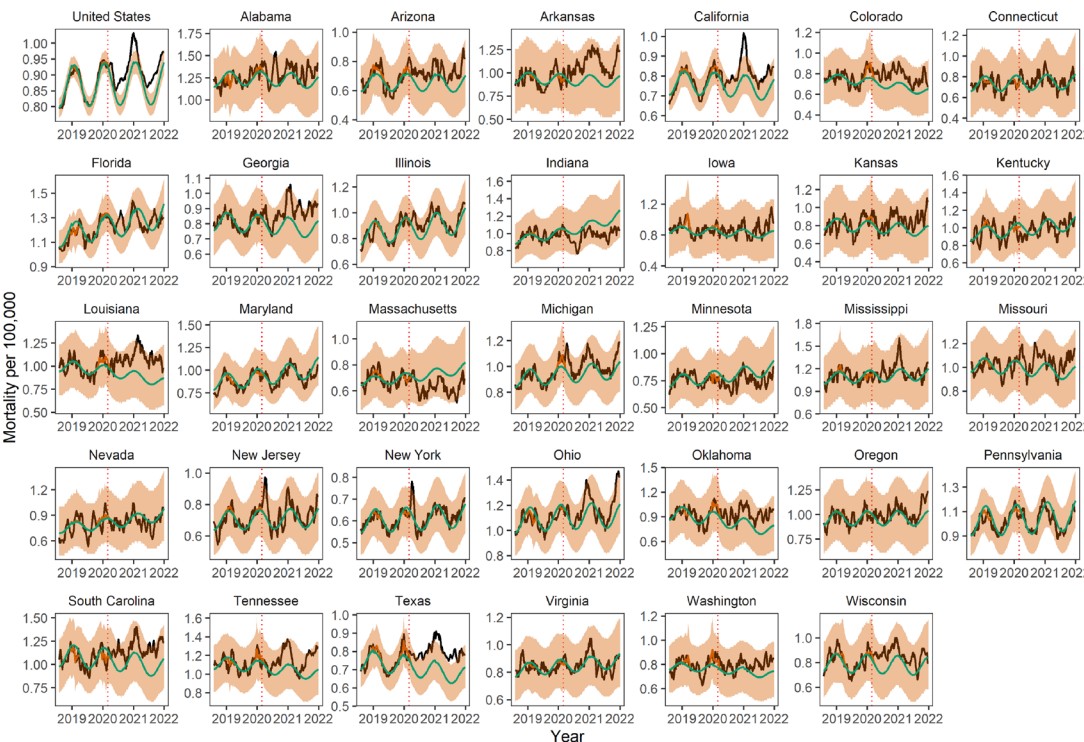

**Appendix 1—figure 4.** Trends in weekly cerebrovascular disease mortality, nationally, and by state. Legend as in *Appendix 1—figure 1*

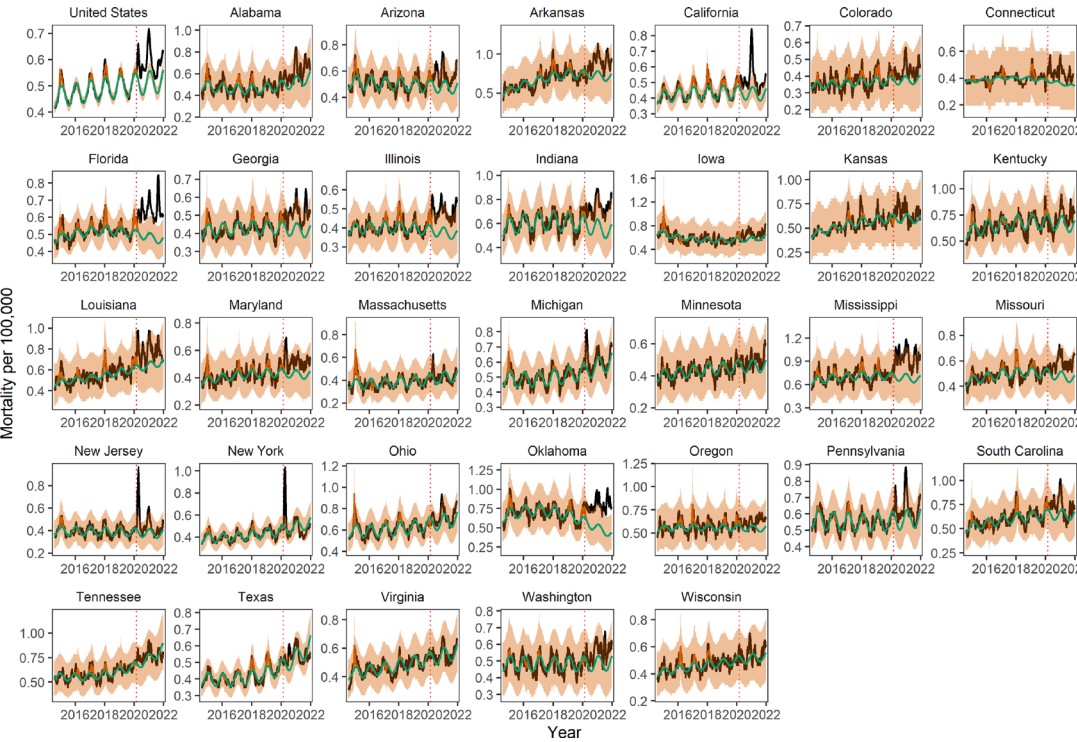

**Appendix 1—figure 5.** Trends in weekly diabetes mortality, nationally, and by state. Legend as in *Appendix 1—figure 1*

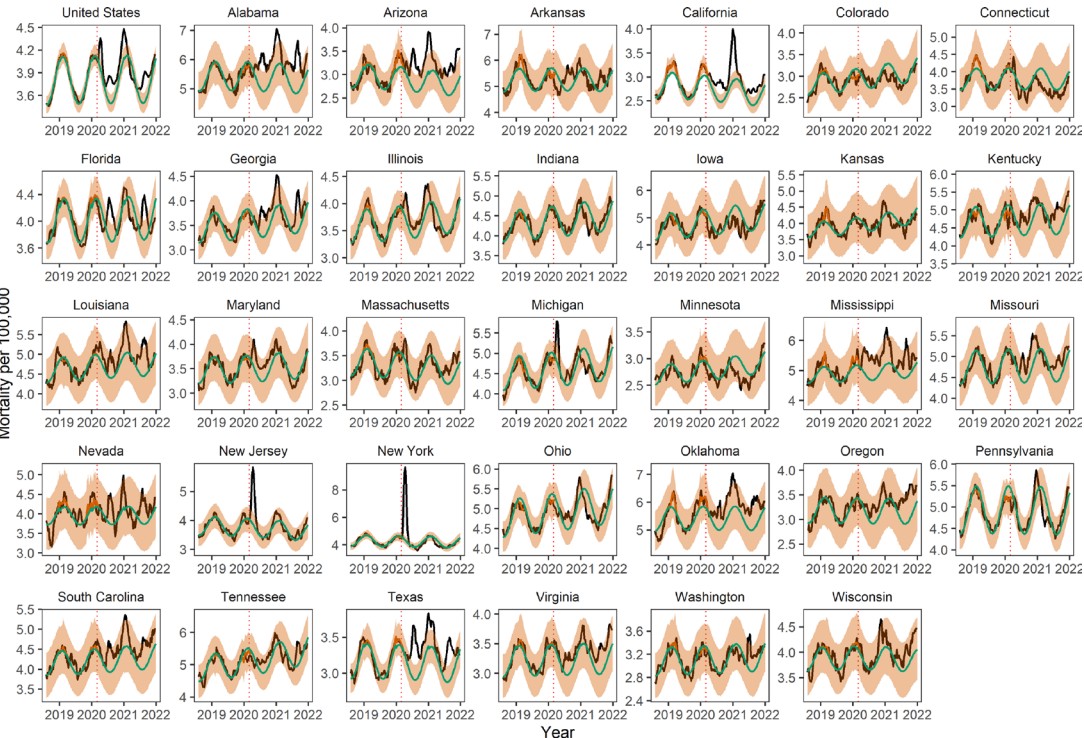

**Appendix 1—figure 6.** Trends in weekly heart disease mortality, nationally, and by state. Legend as in *Appendix 1—figure 1*

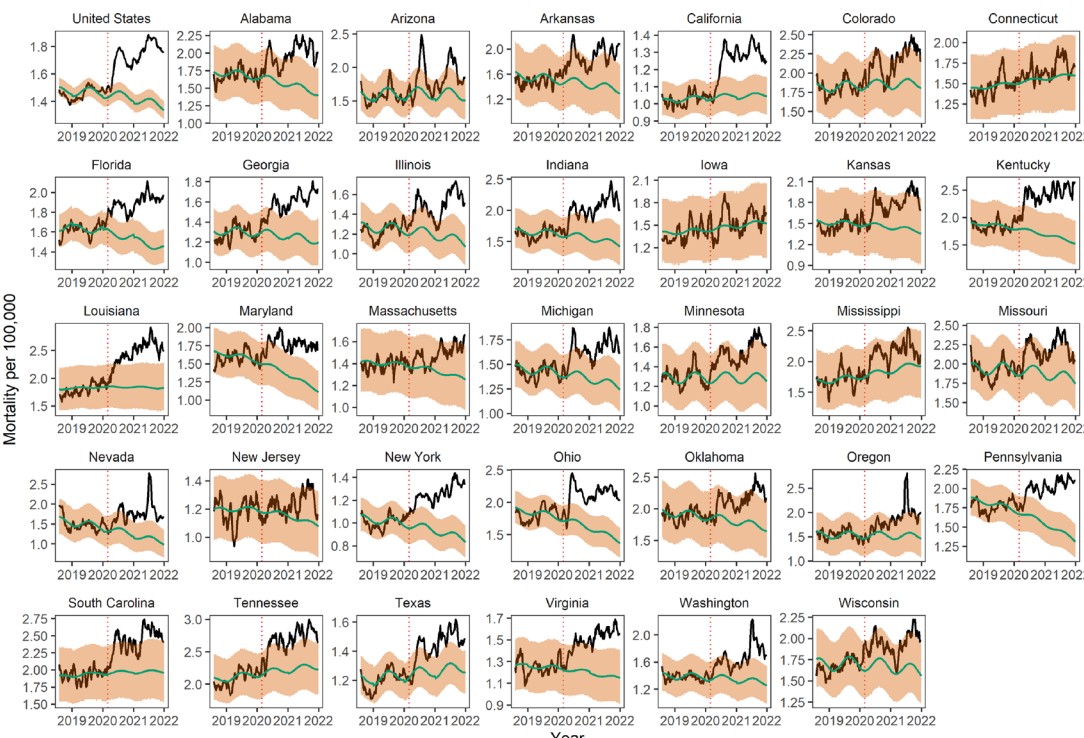

**Appendix 1—figure 7.** Trends in weekly mortality from external causes (opioids, suicides, accidents, etc.), nationally and by state. Legend as in *Appendix 1—figure 1*.

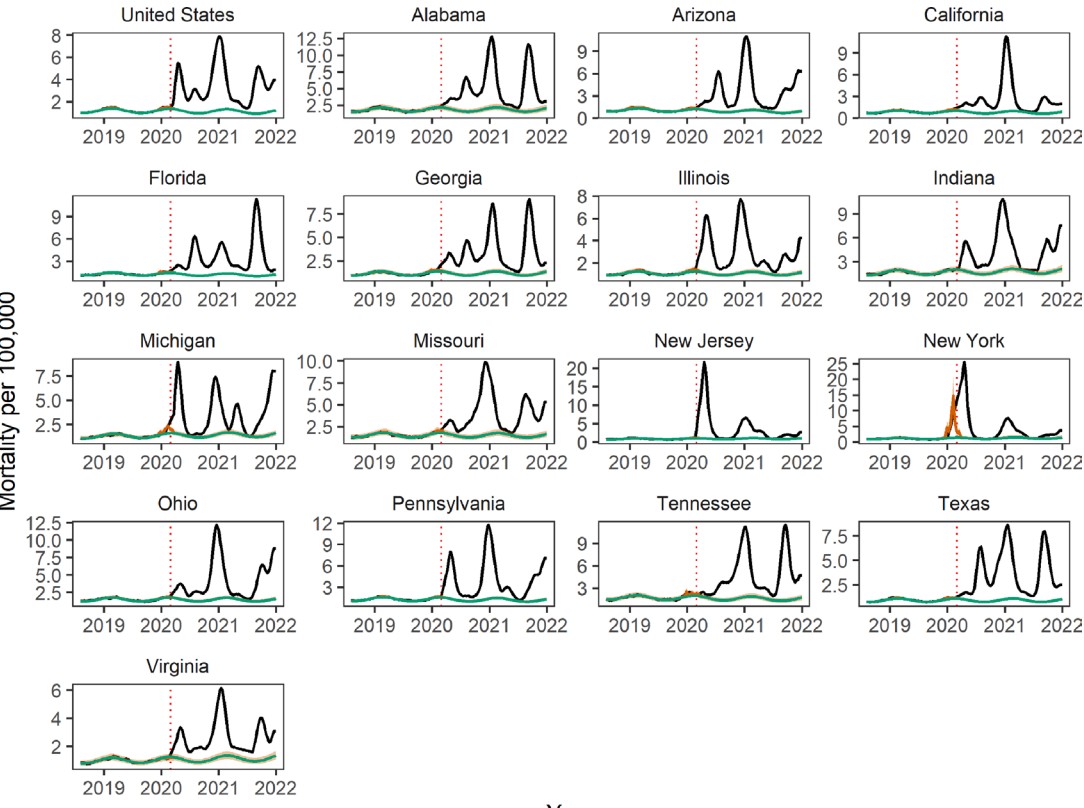

**Appendix 1—figure 8.** Trends in weekly respiratory and COVID-19 mortality, nationally, and by state. Legend as in *Appendix 1—figure 1* (only 16 states had sufficient weekly data for these conditions).

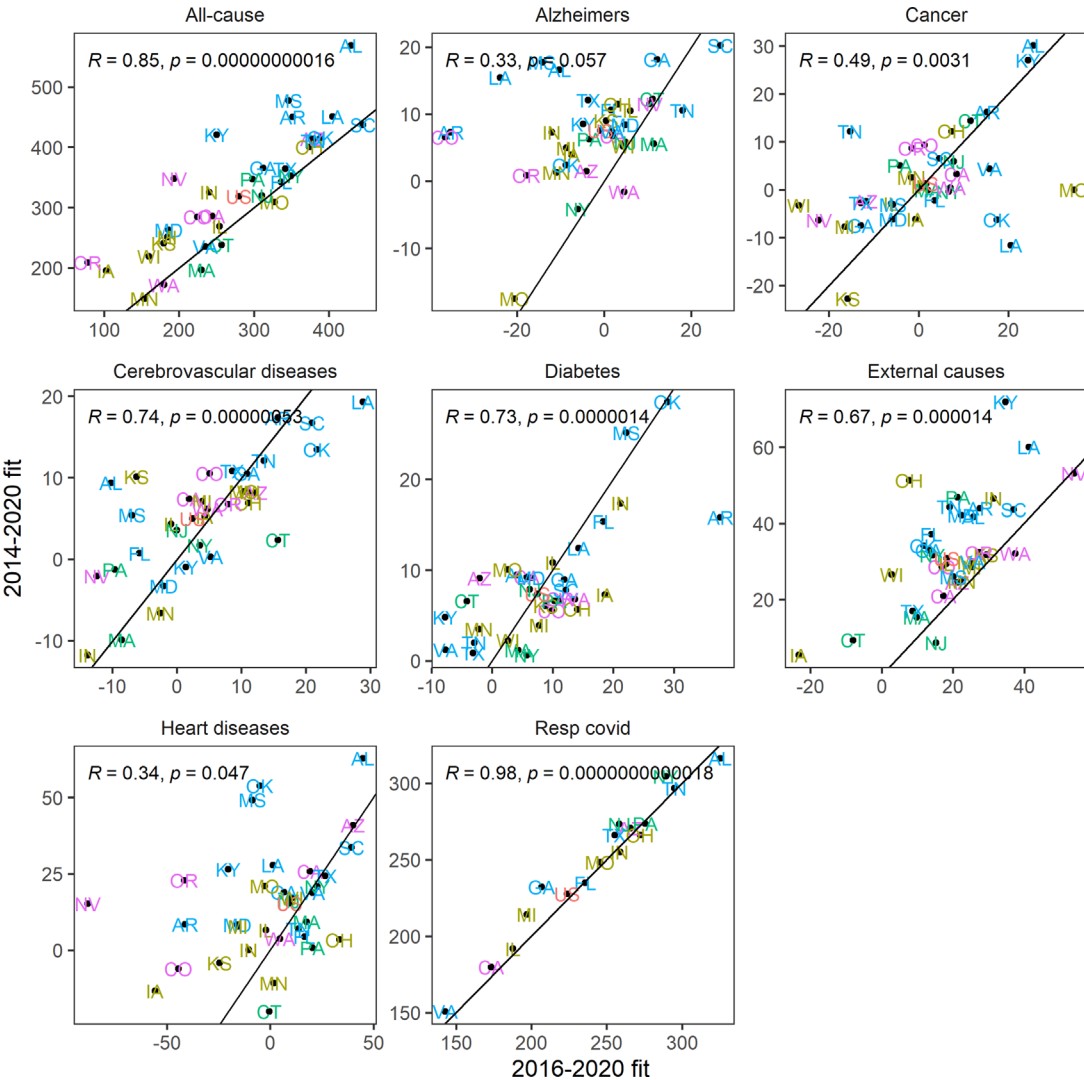

**Appendix 1—figure 9.** Sensitivity analysis using a shorter historic time series (August 1, 2016 to March 1, 2020) than in the main analysis to calibrate the model baseline. Graph displays estimates of cumulative excess mortality rates for the pandemic period, March 1, 2020 to January 1, 2022, by state and cause of death. Correlation is shown with estimates from the main analyses, which use calibration data going back to August 2014.

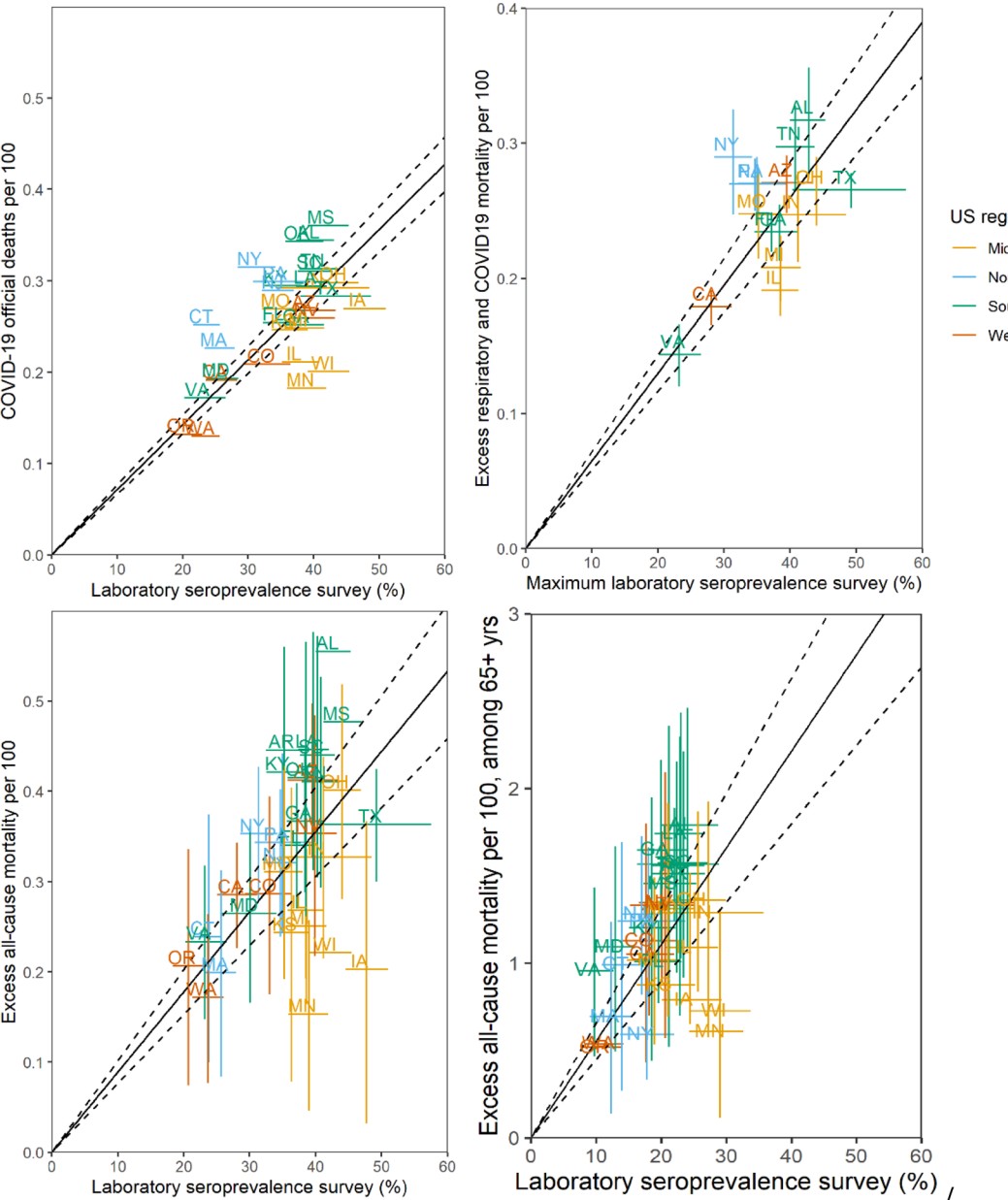

**Appendix 1—figure 10.** Sensitivity Analyses of Infection Fatality Rate (IFR, ratio of deaths/infections), March 1, 2020 to January 1, 2022. The main analysis (*Figure 1B* in main text) is based on a comparison of excess respiratory deaths with Centers for Disease Control and Prevention (CDC) laboratory seroprevalence survey of SARS-CoV-2 N antigen by December 2021. Sensitivity analyses are based on: top left: official COVID-19 death tallies; top right: maximum seroprevalence estimates during study period; bottom left: excess all-cause deaths; bottom right: ages over 65 years (excess all-cause mortality in 65+ vs. seroprevalence in 65+). Each point corresponds to a state, annotated by their acronym. Error bars represent 95% confidence intervals (CIs). The black line and dotted region represent a linear regression fit and the associated 95% CI. IFR estimates, based on the slope of the regression, are: official COVID-19: 0.71% (0.66–0.76%); maximum serology estimates: 0.65% (0.58–0.72%); all-cause mortality 0.90% (0.76–1.01%); 65+ years: 5.5% (4.5–6.6%).

We also ran a sensitivity analysis to test the robustness of the baseline, and resulting excess mortality estimates, to the number of years included in the model. We fitted the baseline model to a shorter historical time series from August 1, 2016 to March 1, 2020, while the main analysis considered 2 additional years of data (2014–2020). National cause-specific excess mortality estimates were robust to these changes, and so were state-specific estimates for all-cause and respiratory and COVID-19 conditions (*Appendix 1—figure 9*). State-specific estimates for certain causes, particularly

Alzheimer's, were more sensitive to the amount of data included in the baseline; however, the ranking of excess mortality across states was robust to the length of data used for model fitting even for Alzheimer's. *Appendix 1—figure 9* shows the impact of using shorter historical time series on cumulative excess mortality estimates for the pandemic period, by state and cause of death.

Of note, the impact of seasonal influenza was estimated by having an explicit flu coefficient in the excess mortality model, while the COVID-19 impact was estimated by taking the difference between the observed mortality and baseline mortality during the pandemic period. Generally, this would be expected to inflate the impact of COVID-19, relative to influenza. It can also be difficult to compare the impact of influenza, which circulates over a relatively short winter season (typically 3 months), while COVID-19 has caused persistent excess mortality over 2 years. To address this issue, we considered a standard period, November through March, to compare the impact of the two pathogens. We estimated excess mortality during the severe influenza season of November 2017 to March 2018, compared to excess mortality during November 2020 to March 2021, a period when COVID-19 mortality was the most widespread in the US. This comparison should be caveated by the fact that our excess mortality estimates of COVID-19 captures direct and indirect pandemic effects, while by construction our influenza estimates represent direct viral effects. Yet, the scale of the difference of excess mortality between the two pathogens (five to sixfold mortality rate ratio across most US states, *Appendix 1—figure 11*) highlights the magnitude of SARS-CoV-2 disease burden.

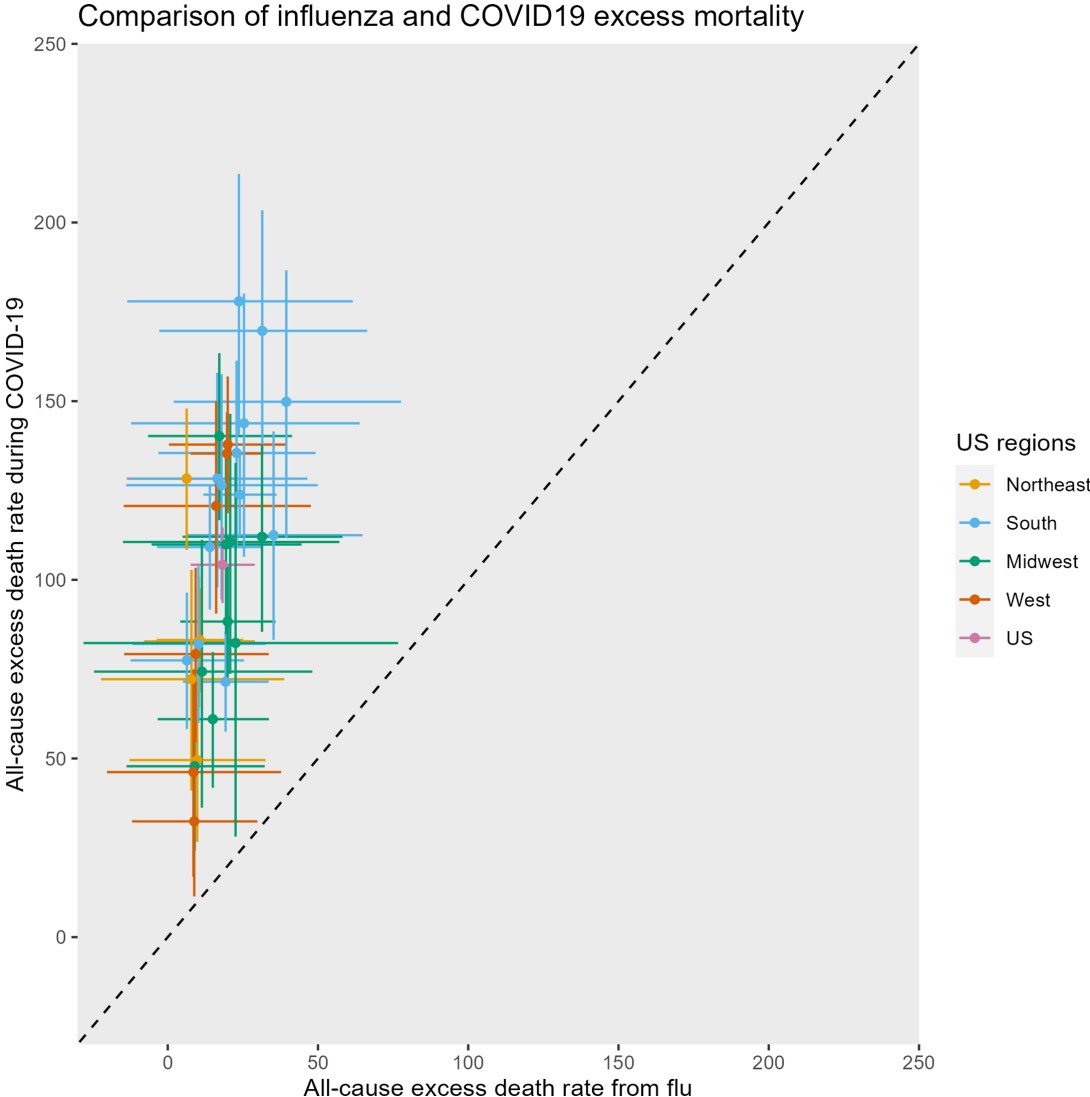

**Appendix 1—figure 11.** Comparison of excess mortality from a severe influenza season (2017–2018) and the most severe of the COVID-19 waves (2020–2021 winter wave) across United States. For the sake of comparability, we estimated all-cause excess mortality for the November tto March period for each pathogen. All rates are per 100,000. Each symbol represents a state, color-coded by region, with the horizontal and vertical bars representing 95% confidence intervals (CIs) on excess estimates. Note that some of the lower bounds of the CIs on the flu estimates were negative. The diagonal represents the line where the influenza and COVID-19 mortality burden would be equal. The median ratio of excess COVID-19 to excess flu is 5.8 across United States.

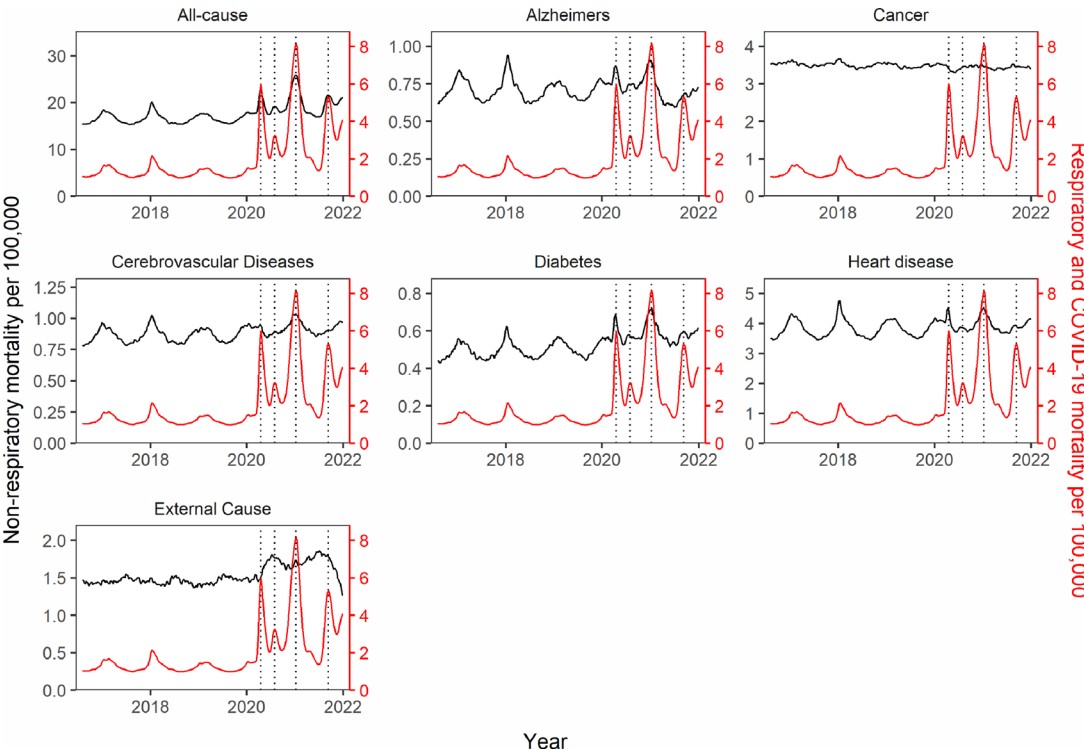

**Appendix 1—figure 12.** Synchrony between respiratory and non-respiratory mortality patterns on a national scale. The black lines show the time-series of mortality rates (before excess mortality modeling) for each non-respiratory mortality cause. The red line represents respiratory and COVID-19 mortality rate. The dotted black vertical lines mark the dates of peak respiratory and COVID-19 mortality for each of the four national pandemic waves in the study period (April 18, 2020; August 1, 2020; January 9, 2021; September 11, 2021). Synchrony with respiratory and COVID-19 mortality is most pronounced for all-cause, Alzheimer's, Heart Disease, Diabetes, and in the first three waves (before vaccination). Synchrony is intermediate for cerebrovascular diseases and absent for cancer and external causes.

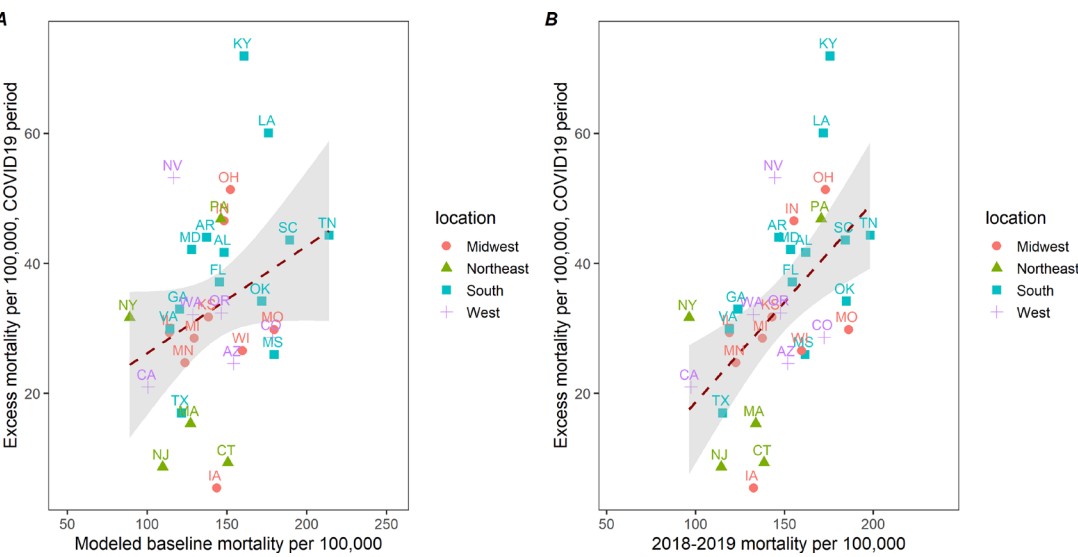

**Appendix 1—figure 13.** Correlation between cumulative excess death rates due to external causes (opioids, suicide, accidents, etc.) during March 1, 2020 to January 1, 2022 and baseline death rates of external causes, across 33 states. (**A**) Baselinedeath rates from external causes are based on the seasonal regression model shown in *Figure 3*. (**B**) Baselines death rates are based on observed mortality rates from external causes for a comparable pre-pandemic period, March 2018 to December 2020. The correlation is non-significant on the left (corr=0.30, p=0.09) and moderate on the right (corr=0.54, p=0.001). This suggests that states that had high rates of mortality

*Appendix 1—figure 13 continued on next page*

from external causes in the 2 years before COVID-19 saw a more pronounced mortality elevation during the pandemic.

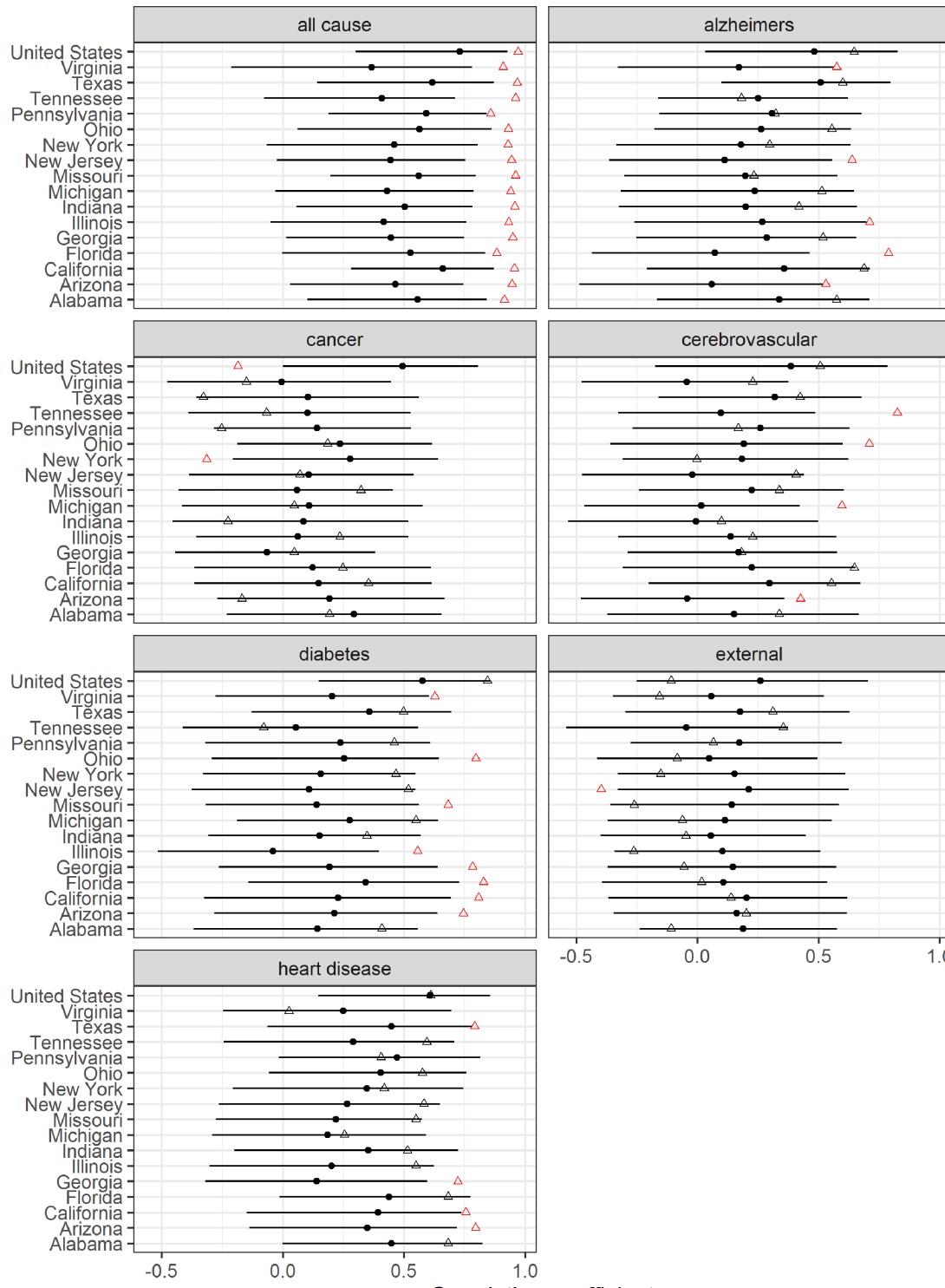

**Appendix 1—figure 14.** Changes in weekly synchrony between respiratory and non-respiratory mortality causes during the pandemic. Graph compares correlations during 96 weeks of any baseline pre-pandemic period and in the 96 weeks of the pandemic (March 2020 1 to January 1, 2022). Black points represent estimated pre-pandemic correlations (96 weeks selected before March 2020 by block of 2 weeks). Black error bars represent 95% bootstrap

*Appendix 1—figure 14 continued*
confidence intervals (CI) accounting for multiple comparisons using Bonferroni correction. Triangles represent estimated pandemic correlations. Red color indicates significant deviation from pre-pandemic correlation. Correlation is highest for all-cause and is more pronounced during the pandemic period (red triangle), which suggests a direct impact of the virus on these conditions.

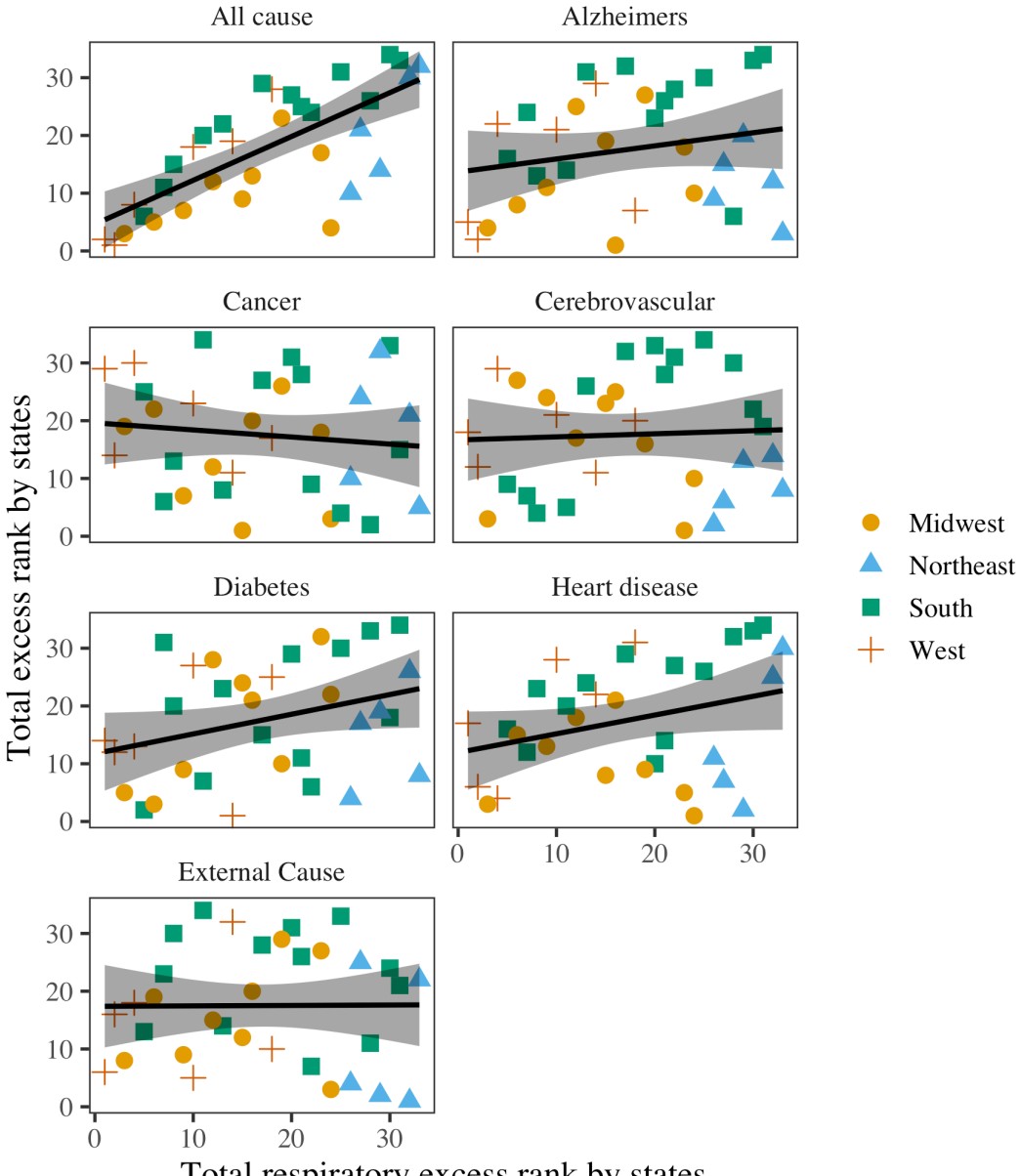

**Appendix 1—figure 15.** Rank correlation between total COVID-19 mortality and total excess mortality for other causes, across 33 states. Black lines represent the best fit regression lines. Shaded areas represent the 95% confidence intervals (CI). The states have been categorized into the Midwest, Northeast, South, and West. Respiratory deaths are moderately to highly correlated with all-cause (rho=0.73, 95% CI: 0.47–0.90).

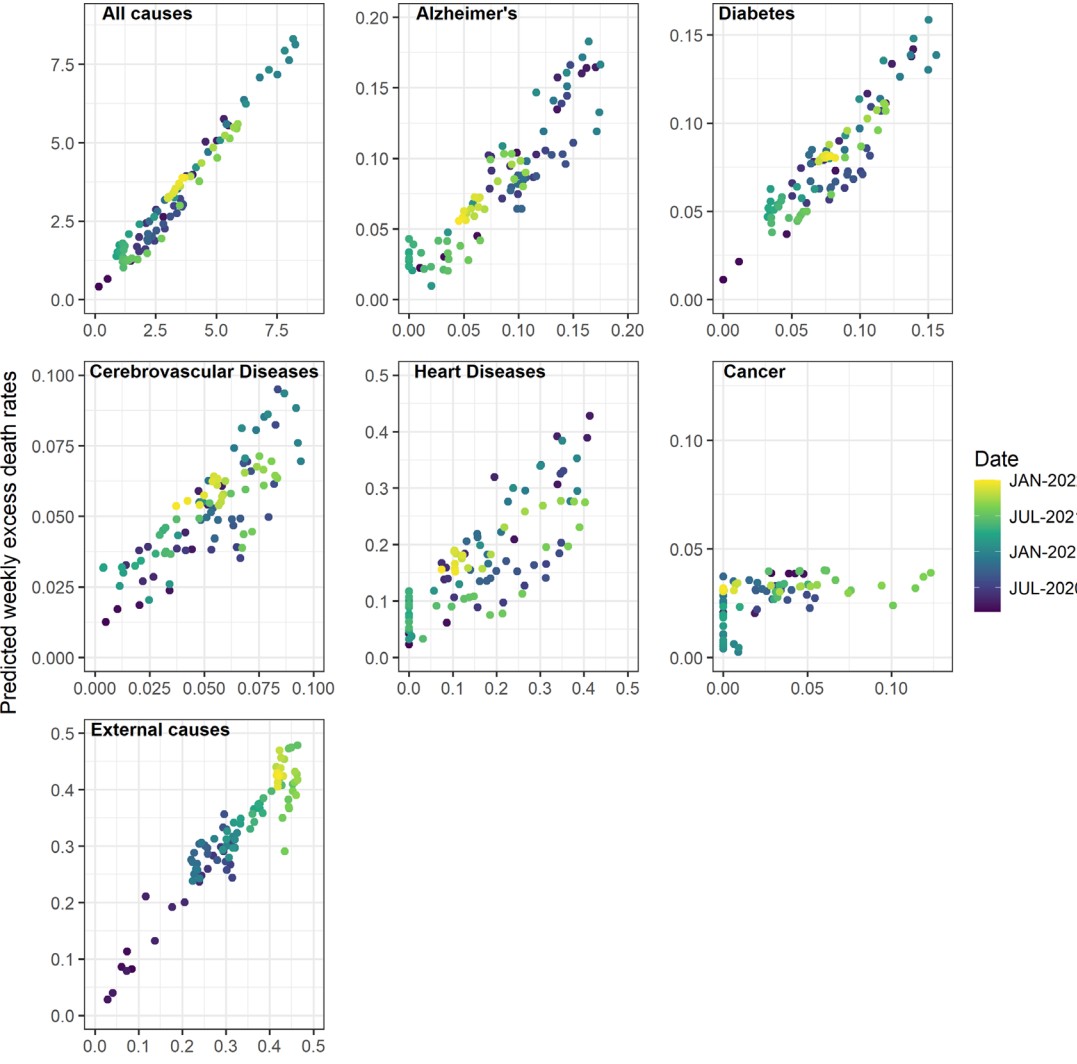

**Appendix 1—figure 16.** Observed and predicted excess death rates by condition, United States, March 1, 2020 to January 1, 2022, using Generalized Additive Model (GAM) models with weekly COVID-19 intensity, intensive care unit (ICU) occupancy, and strength of interventions as covariates. Observed values are on the x-axis and predicted on the y-axis, with colors representing different time periods.

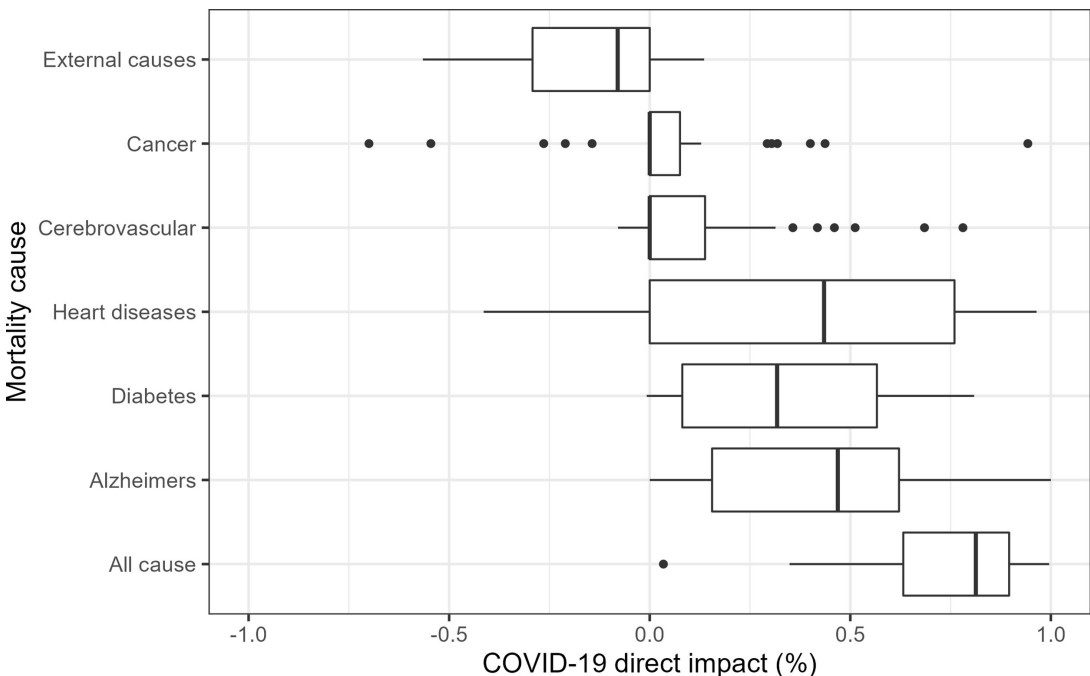

**Appendix 1—figure 17.** State-level analyses of the direct contribution of COVID-19 on weekly mortality from different causes, March 1, 2020 to January 1, 2022. Results as based on Generalized Additive Model (GAM) with weekly excess mortality as the outcome, and weekly COVID-19 intensity, intensive care unit (ICU) occupancy and strength of interventions as covariates. Models are fit separately to each state. The box plots represent the proportions of excess deaths attributed to COVID-19 across states (i.e. the direct impact of SARS-CoV-2 infection). These proportions can be compared to national estimates displayed in *Table 2*. As in national data, we see that the state-level contribution of COVID-19 on all-cause, Alzheimers, diabetes, and heart diseases is substantial. Furthermore, the contribution to external causes is typically negative, indicating that excess mortality from these conditions tends to be high when COVID-19 activity is low. There is no or low contribution to cancer and cerebrovascular diseases.

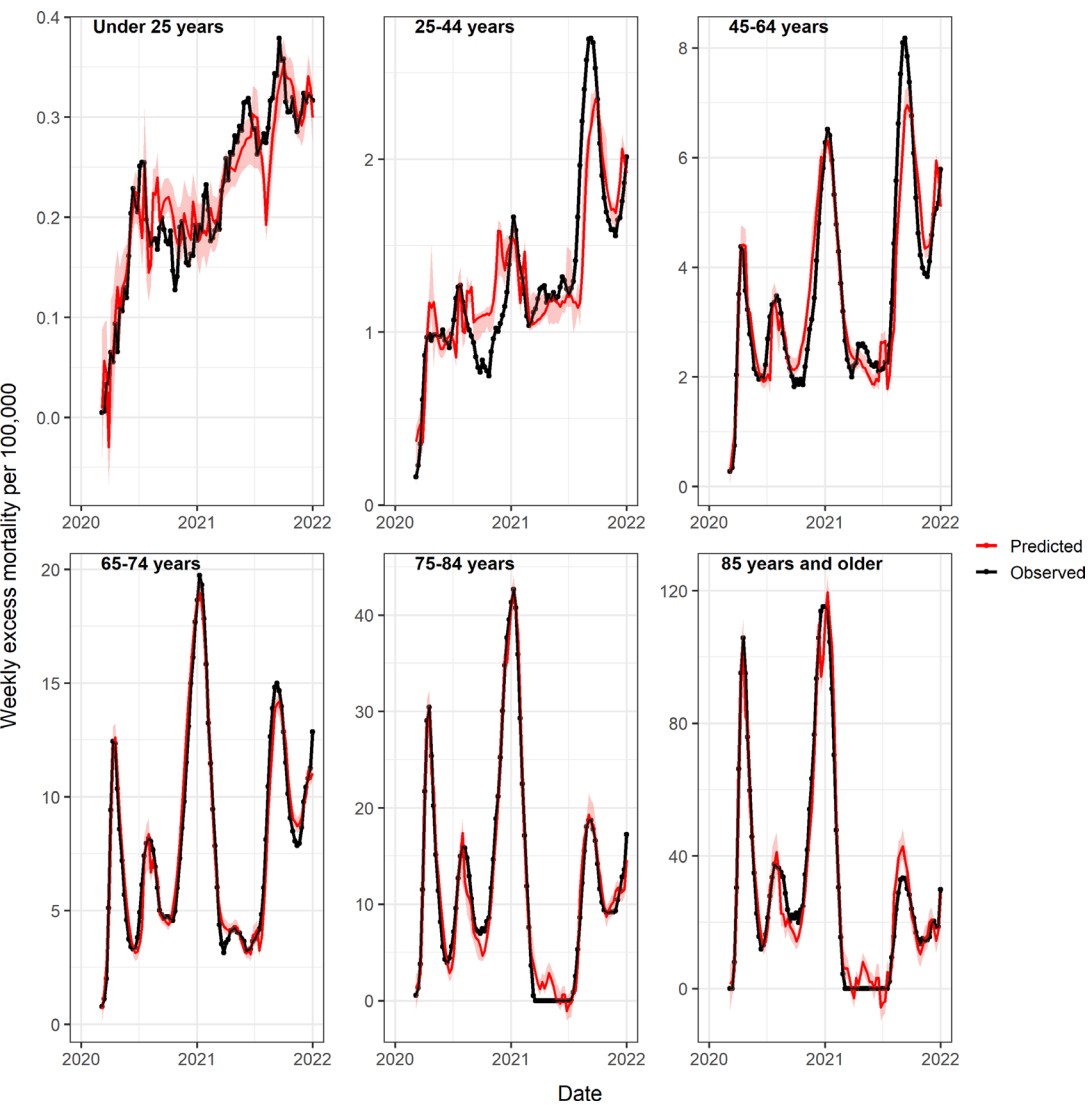

**Appendix 1—figure 18.** Observed and predicted excess death rates by age group, United States, March 1, 2020 to January 1, 2022, using Generalized Additive Model (GAM) models with weekly COVID-19 intensity, intensive care unit (ICU) occupancy and strength of interventions as covariates. Observed values are in black and predicted values in red (mean=dark red, 95% CI in lighter red).

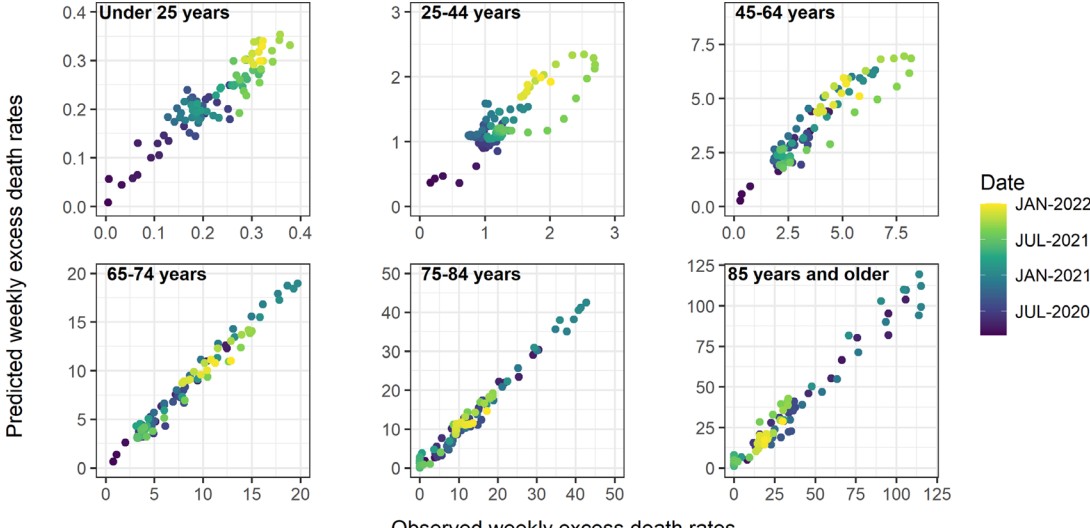

**Appendix 1—figure 19.** Observed and predicted excess death rates by age group, United States, March 1, 2020 to January 1, 2022, using Generalized Additive Model (GAM) models with weekly COVID-19 intensity, intensive care unit (ICU) occupancy, and interventions as covariates. Observed values are on the x-axis and predicted on the y-axis, with colors representing different time periods.

## Models based on monthly excess mortality model for subcategories of external deaths

To better understand the rise in external deaths during the pandemic, we applied a similar excess mortality approach to deaths from five subcategories of external deaths available at a monthly resolution (*Centers for Disease Control and Prevention, 2022d*; *Centers for Disease Control and Prevention, 2022e*): all accidents, motor vehicle accidents, drug overdoses, assaults and homicides and suicides. Because these data were on a monthly time scale, and less stationary that the other mortality causes, we tried more flexible model formulations including spline terms for time trends and/or for seasonality. Based on AIC, a model with cubic spline terms for time trends and seasonality (5 df/year) provided the best fit to the data. We did not include a term for influenza as there is no biological reason for why influenza would affect external deaths. External cause of death data were not available by subcategory and age, but we had age-specific data for all external causes of death combined (*Centers for Disease Control and Prevention, 2022e*). We visually explored these age-specific data as there was not enough information to fit time series models (*Appendix 1—figure 20*).

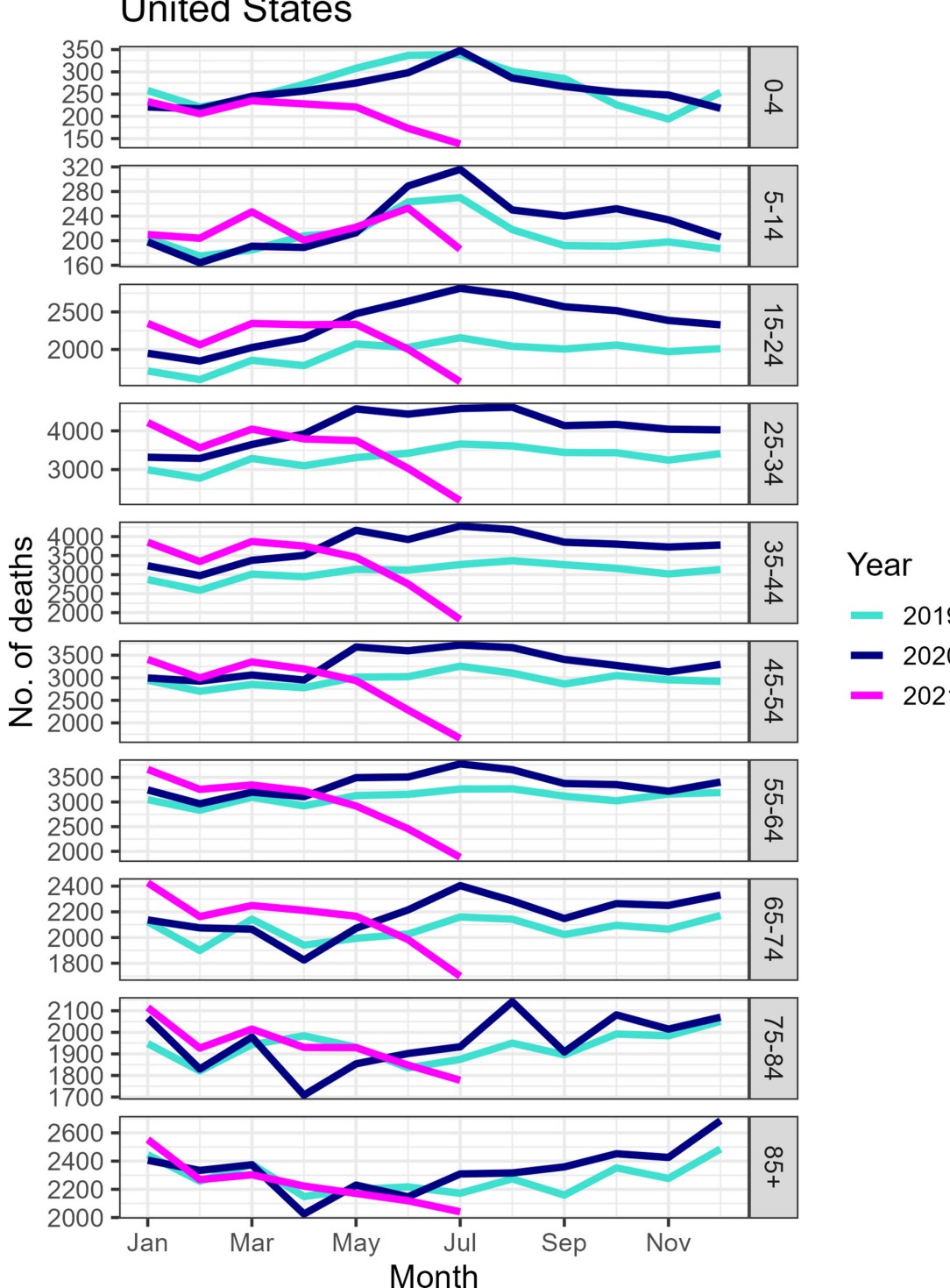

**Appendix 1—figure 20.** Monthly number of deaths from external causes by age group, United States, January 2019 to July 2021. Although mortality appears to have decreased substantially by June to July 2021, some of the decrease is likely attributable to reporting delays as the underlying dataset does not appear to have been updated recently.

## Estimation of the direct and indirect mortality impacts of the pandemic

We use a two-step approach to estimate the direct and indirect impacts of the pandemic. In the first step, we estimate national weekly excess mortality for a given cause of death and age group, along

with CI, based on the model detailed in the prior section. In a second step, we regress weekly cause- and age-specific excess mortality against a weekly proxy for the strength of non-pharmaceutical interventions Oxford Health containment index (**Oxford University, 2021**), weekly hospital strain (from HHS hospital occupancy mandatory reporting of ICU use during COVID-19 **Health and Human Services, 2022**), and weekly COVID-19 activity (proxied by the official tallies of COVID-19 deaths in NCHS data). To allow for non-linear effects between covariates and excess mortality, we used GAM, with each covariate modeled as a smoothing spline (mgcv package in R). All models have intercepts to account for the impact of factors not captured by our three covariates.

To derive the direct contribution of COVID-19 to excess mortality, we set the COVID term to zero and calculate the ratio of (predicted_deaths_full_model – predicted_deaths_0_COVID)/ predicted_deaths_full_model. This can be understood as the attributable fraction of COVID-19 on excess deaths, for any cause. We explored potential delays between excess mortality and covariates (intervention measures, hospital strain, and COVID-19 activity) using cross-correlation analysis (ccf function in R). We identified a lag of 4 weeks for interventions, 5 weeks for hospital strains, and no lag for COVID-19, consistent across mortality outcomes and age groups.

To propagate the uncertainty in excess mortality estimates (response variable) between the first and second regression steps, we resampled the weekly excess mortality estimates based on their mean estimated values and standard deviation provided by the step 1 seasonal regression model, assuming a normal distribution. We sampled excess mortality 1000 times at each week (generating 1000 time series of excess mortality) and performed GAM regression using COVID-19 activity, hospital strain and interventions as covariates. Then, for each of the 1000 regressions, we approximated the estimated direct contribution of COVID-19 as in the analysis of main data. We estimated the CI for the effect of COVID-19 on mortality based on the quantiles of the simulated estimates. This method accounts for uncertainty in the response variable as well as uncertainty in each GAM model fit. The same approach was used for national cause- and age-specific data, and for state-level data.

An alternative approach to our two-step process (Step 1: estimate weekly excess mortality and Step 2: regress weekly excess mortality against weekly COVID-19 activity, hospital use, and weekly interventions) would be to run a single regression model, where observed weekly mortality rates are regressed against seasonal terms, time trends, flu activity, COVID-19 activity, hospital use, and interventions. However, because there were only 96 pandemic weeks in our dataset, compared to 292 pre-pandemic weeks where the COVID-19 and intervention coefficients were zero, and hospital ICU use data was unavailable before the pandemic, this alternative approach would be inappropriate to get a robust estimate of the direct and indirect pandemic impacts.

## Estimation of the IFR

Without accounting for delays between infection and death, the IFR can be estimated by dividing the total number of deaths by the total number of infections. In other words, we have:

$$IFR = \frac{Total\ number\ of\ deaths}{Total\ number\ of\ infections},$$
$$= \frac{(Total\ number\ of\ deaths)/(Population\ size)}{(Total\ number\ of\ infections)/(Population\ size)},$$
$$= \frac{Excess\ respiratory\ deaths\ per\ 100\ people}{\%\ seroprevalence}$$

assuming that excess respiratory deaths provide an accurate estimate of deaths attributable to COVID-19. If we have information on excess deaths and serology in multiple states, we can regress these factors against each other, and the slope gives the average nationwide IFR when the intercept is set to zero (**Figure 1**). Given that the delay between seroconversion after infection is around 2 weeks, and the delay between infection and death is in the same order of magnitude, we use excess deaths until January 1, 2022 in the numerator and serology for the last week of December 2021 in the denominator.

To propagate uncertainty from both the response variable (excess mortality) and covariate (seroprevalence) into IFR estimates, we used a similar approach as for the direct and indirect attribution model in the previous section. We resampled from the reported estimates of excess mortality and seroprevalence in each state, assuming normal distributions aligned with the reported 95% CI. Then, we drew 10,000 samples of excess mortality and seroprevalence in each state, and for each sample data set, we performed a linear regression and estimated the slope. Next, we drew 100 random samples for each of the 10,000 slope distributions. We calculated the CI of our IFR estimate

by aggregating random samples of IFR estimates across all 100,000 sample data sets and taking 2.5 and 97.5% quantiles.

Different sensitivity analyses on the IFR are presented in *Appendix 1—figure 10*.

