## [Editor Report]

The authors examine the impacts of the COVID-19 pandemic on excess mortality in the US up to January 2022. The authors separate direct impacts of the pandemic from indirect impacts (disruptions), finding that most excess deaths (84%) are due to direct impacts. Moreover, in individuals under 44 years of age, indirect effects predominate in mortality from external causes and all-cause mortality. The paper is well written and of interest to understant the impacts of the COVID-19 pandemic.

---

## [Decision Letter]

**Decision letter after peer review:**

Thank you for submitting your article "Direct and indirect mortality impacts of the COVID-19 pandemic in the US, March 2020-April 2021" for consideration by *eLife*. Your article has been reviewed by 3 peer reviewers, one of whom is a member of our Board of Reviewing Editors, and the evaluation has been overseen by David Serwadda as the Senior Editor. The reviewers have opted to remain anonymous.

Essential revisions:

This is a very interesting paper quantifying excess deaths due to the COVID-19 pandemic in the USA. The paper is roughly divided into two main sections. In the first section, the authors estimate age and cause-specific excess mortality. In the second section, using their excess mortality estimates, the authors attempt to disentangle the impact of SARS-CoV-2 infection (direct impact) vs. the impact of NPIs on this excess mortality (indirect impact). While all reviewers agree that this is a relevant contribution, the paper needs some revisions/clarifications, particularly in the second section.

1. We would encourage authors to discuss the two different concepts of excess mortality:

(#1) what deaths were caused, directly or indirectly, by the pandemic. This is what authors seem to have aimed to assess (#2) how many additional deaths occurred during the pandemic, compared to what would have been expected in the absence of a pandemic. For such an analysis I think expected annual influenza deaths should be added back to the baseline (or subtracted from the excess)? Some of the discussion seems to relate more to an impression of #2 rather than #1 so it would be good to be explicit about what is being estimated.

2. Choosing the baseline/counterfactual is a key part of excess mortality analyses. I've seen other papers exploring different potential periods (shorter periods; either by using older or newer years, e,g 2017-2020 or 2014-2017) to test sensitivity to the choice of these periods. Moreover, I would think that an absence of other point-in-time disasters would also be a needed assumption (as these would not represent a baseline). It may be beneficial for the reader to outline explicitly the assumptions of their model. Some of these assumptions are outlined on page 5, but I am not sure whether this is an extensive list of the assumptions of this approach.

3. There are no mentions in the description of mortality data of two common issues with vital registration data. First, the presence of ill-defined deaths (R chapter, and when subdividing injuries, Y10-Y34). If these increase or decrease during the pandemic because of differentials in coding, it may over or underestimate the presence of excess mortality. Second, the lack of coverage of all deaths. Helleringer talks a bit about that here https://academic.oup.com/ije/article/51/1/85/6460626?login=true in the context of estimating excess mortality. While this issue may seem trivial in the US, Murray shows here that there is a lack of complete coverage that is differential by county in the US https://journals.plos.org/plosmedicine/article?id=10.1371/journal.pmed.1000262. While this may or may not affect the authors' analysis, I think the implicit assumption of no changes in the coverage of deaths should be made explicit.

4. We are concerned about the analysis estimating the impact of NPIs. The authors are using explicit causal language throughout this section, but there's very little discussion about the methods used and the assumptions and limitations of these methods. Specifically:

a. Please provide a bit more information on the model used for direct vs. indirect effects. We'd like to see explicit discussion of the assumptions and limitations of the approach but also of the stringency index used. Was the association between stringency index and excess deaths assumed to be linear? Or were different functional forms considered? It is also not clear how well the model fit the data.

b. Related to the above, please provide more details on how the results of the regressions were translated into the results presented. The main text reports percentages, but the methods only briefly explain how numbers of direct deaths were calculated, and the supplementary tables report coefficients. It is not clear if these estimates of direct and indirect deaths were somehow constrained to add up to the total number of excess deaths, but it doesn't seem like it since point estimates cross 100% in some cases.

c. Please be more explicit about causal assumptions (see comment 6 of reviewer 2).

d. Please discuss the potential limitations of using the stringency index to quantify NPIs particularly since it doesn't take into account the actual implementation of the measures (and how this may have varied over time).

*Reviewer #2 (Recommendations for the authors):*

In this paper, the authors examine the impacts of the COVID-19 pandemic on excess mortality in the US up to April 30, 2021. The authors separate direct impacts (caused by COVID-19, coded as such or not) of the pandemic from indirect impacts (disruptions), finding that most excess deaths (90%) are due to direct impacts. Importantly, the authors find that the official COVID-19 death tally is an undercount of these deaths. Moreover, the authors also find that excess deaths due to other causes are the main driver of excess mortality among younger populations. This is a very important research question that the authors have addressed using robust methods. I have a few suggestions that I hope can help clarify the reporting of results and outlining of assumptions.

1. Abstract: since the study presents several axes of comparison (old vs young, direct vs indirect, specific causes vs one another) the abstract is quite complicated, and there are a few sections that may benefit from clarification. Specifically: (a) the authors mention the indirect effects

2. The framing of the study in the introduction may benefit from a more comprehensive outlining of potential directions that mortality may change with the pandemic. For example, declines in mortality have been observed in many settings (e.g. Chile and Guatemala; or in Figure 3 of this paper) following lockdowns in areas with low COVID-19 spread. This is mostly due to declines in external causes (mostly traffic crashes) and air pollution-related deaths (e.g. CVD or infant deaths). The authors just mention one potential directionality (COVID-19 increasing some deaths directly, and NPIs causing other deaths) without any mention to a lowering of mortality that may obscure these increases.

3. Choosing the baseline/counterfactual is a key part of excess mortality analyses. I've seen other papers exploring different potential periods (shorter periods; either by using older or newer years, e,g 2017-2020 or 2014-2017) to test sensitivity to the choice of these periods. Moreover, I would think that an absence of other point-in-time disasters would also be a needed assumption (as these would not represent a baseline). It may be beneficial for the reader to outline explicitly the assumptions of their model. Some of these assumptions are outlined on page 5, but I am not sure whether this is an extensive list of the assumptions of this approach.

4. I see no mentions in the description of mortality data of two common issues with vital registration data. First, the presence of ill-defined deaths (R chapter, and when subdividing injuries, Y10-Y34). If these increase or decrease during the pandemic because of differentials in coding, it may over or underestimate the presence of excess mortality. Second, the lack of coverage of all deaths. Helleringer talks a bit about that here https://academic.oup.com/ije/article/51/1/85/6460626?login=true in the context of estimating excess mortality. While this issue may seem trivial in the US, Murray shows here that there is a lack of complete coverage that is differential by county in the US https://journals.plos.org/plosmedicine/article?id=10.1371/journal.pmed.1000262. While this may or may not affect the authors' analysis, I think the implicit assumption of no changes in the coverage of deaths should be made explicit.

5. Direct deaths: the authors use COVID-19, influenza, pneumonia, and COPD deaths as an indicator of respiratory mortality. Is this the list of ICD codes in P28 (2.a) of the appendix? The authors also use "mention of COVID-19" as an indicator of directly caused deaths. How is this operationalized? Is it ICD10 code U07? Or does this not rely on ICD coding? The list of respiratory codes in the supplement makes it seem like this is ICD based, but it is unclear from the text itself.

6. I am concerned about the analysis estimating the impact of NPIs. The authors are using explicit causal language throughout this section, but there's very little discussion about the causal assumptions underpinning this analysis. Specifically: (a) I see no approach to addressing lack of exchangeability (for example, areas that implemented more NPIs may have a more vulnerable population, hence the political will to implement them); (b) no discussion of issues with levels of analysis from a jurisdictional point of view, since the level of devolvement of power from states to local jurisdictions varies widely by state, making state-level analysis potentially tricky (and county-level ones too for that matter); and (c) issues with measurement error, specifically differential measurement error of the outcome, influenced by the exposure (improved vital registration coding in areas with more or less NPIs due to better established public health systems). Last, the authors are framing this analysis as a way to "quantify the direct and indirect impacts of the pandemic on different causes of death". I am failing to grasp the intuition of how this analysis helps with this objective, so it may need to be explicitly set in the methods section.

7. I want to commend the authors for their presentation of results. Specifically, the rounding of mortality estimates to avoid false notions of precision and very clear figures. A few notes on figures: (a) figure 1 is lacking a title in my PDF, and the black line is not included in the legend I am guessing it corresponds to the state in the facet title; (b) figure 2 is missing confidence intervals for respiratory illnesses; (c) I suggest the authors map the results of table 1, as a spatial pattern will most likely emerge (top by excess mortality are all Deep South or Northeast states).

8. The results comparing COVID-19 to the 2017/2018 flu season are very interesting, as they help contextualize the impact. However, I am a bit concerned about "denominators" here, specifically time. The authors compared 13 months of excess mortality to an undetermined number of months (4? 6? 12?). I am not sure how influenza seasons are conceptualized, and I think it'd be good to be explicit with its duration. I understand that the authors may be wanting to compare a "full season" of influenza vs one of COVID-19, but is this direct comparison even possible, considering the different (or lack of?) seasonality of COVID-19?

9. There's another result I am having trouble (a) interpreting it and (b) understanding where it comes from: % of deaths directly attributable to COVID-19. The authors seem to be interpreting this as direct effects of COVID-19 infection, as opposed to indirect effects due to disruptions, at least according to the appendix. My comment 2 above already covers the idea that only thinking about increases in mortality due to NPIs is an incomplete framing, as there are reasons why we expect deaths to actually decline during lockdowns (traffic or air pollution-related ones). There's a cursory mention of this in the appendix, but it is restricted to other respiratory deaths. To follow the authors' example in the appendix, a person with diabetes may also not have died because the air pollution levels in their city plummeted and the MI they were about to have did not happen. Or because their car didn't crash (or another car didn't crash on them) because traffic levels plummeted too. On (b), understanding where it comes from, the appendix provides extra clues (the main text doesn't explain this part that much). If I am understanding things right, the authors are regressing age and cause-specific mortality on NPI strength and on weekly deaths. They are then multiplying the COVID-19 deaths coefficient by the number of deaths to get the # of deaths directly caused by COVID-19. The authors do test potential lags, as both NPIs and deaths may have lagged effects or may be the result of lags between infections and deaths. However, I don't fully follow: (a) why the NPI adjustment is needed at all, (b) why are deaths an adequate COVID-19 activity indicator, especially since authors find a 0-week lag (pretty self-fulfilling, given that they are regressing deaths on a subset of deaths), which doesn't speak much about healthcare disruptions (which occur weeks before that point). One last concern I have is of competing risks, and whether areas with more strict NPIs would have lower direct excess mortality, increasing the % of deaths that are indirectly attributable to the pandemic, not because these deaths were caused by the pandemic, but because they could not have been caused by a well-controlled epidemic (given NPIs).

---

## [Author Response]

Essential revisions:This is a very interesting paper quantifying excess deaths due to the COVID-19 pandemic in the USA. The paper is roughly divided into two main sections. In the first section, the authors estimate age and cause-specific excess mortality. In the second section, using their excess mortality estimates, the authors attempt to disentangle the impact of SARS-CoV-2 infection (direct impact) vs. the impact of NPIs on this excess mortality (indirect impact). While all reviewers agree that this is a relevant contribution, the paper needs some revisions/clarifications, particularly in the second section.

We thank the editor and reviewers for their constructive comments. We acknowledge that the second section needed some clarifications and improvement with respect to the hypotheses tested. We have revised the paper accordingly and added more detailed explanations to the introduction, methods and supplement. We have also updated our models (second section) to include non-linear effects between excess mortality and covariates and included the potential effect of increased hospital occupancy use during COVID-19. Further, we have extended the analysis until January 1, 2022, adding 8 months of data (which is effectively adding the Δ variant wave to the initial analysis). We ran sensitivity analyses on the length of the data used for calibration of the model baseline. We have also revised the comparison between COVID19 and influenza. As a result, the paper has been substantially rewritten and overhauled. These changes have strengthened our analyses, but they do not have any bearing on our conclusions.

1. We would encourage authors to discuss the two different concepts of excess mortality:(#1) what deaths were caused, directly or indirectly, by the pandemic. This is what authors seem to have aimed to assess (#2) how many additional deaths occurred during the pandemic, compared to what would have been expected in the absence of a pandemic. For such an analysis I think expected annual influenza deaths should be added back to the baseline (or subtracted from the excess)? Some of the discussion seems to relate more to an impression of #2 rather than #1 so it would be good to be explicit about what is being estimated.

Thanks for the comment. We have clarified our assumptions and justification for the construction of the baseline. Overall, our excess mortality approach estimates deaths above mortality from past years; here we single out influenza by adding a covariate for flu in the regression and take it out of the baseline. Excess deaths due to COVID19 are deaths above the baseline estimated from prior years, after the impact of influenza has been removed. Our choice is motivated by the large variation in influenza mortality burden between years due to differences in circulating strains – it is not straightforward to define an average flu season. On the other hand, we do not single out deaths from many other viral infections that are presumably less variable between years (for which we do not have viral activity proxies) – these viruses have been greatly reduced during the pandemic and hence this means our baseline is likely inflated.

We have now clarified our assumptions about the baseline in the methods section (a slightly condensed version of the above response is provided in the first paragraph of analytic approach, p 5). Also, we return to the meaning of the baseline in discussion (limitations section, bottom of p12). Further, we now provide a comparison of influenza and COVID-19 excess mortality for a comparable period of virus circulation (bottom of p7).

2. Choosing the baseline/counterfactual is a key part of excess mortality analyses. I've seen other papers exploring different potential periods (shorter periods; either by using older or newer years, e,g 2017-2020 or 2014-2017) to test sensitivity to the choice of these periods. Moreover, I would think that an absence of other point-in-time disasters would also be a needed assumption (as these would not represent a baseline). It may be beneficial for the reader to outline explicitly the assumptions of their model. Some of these assumptions are outlined on page 5, but I am not sure whether this is an extensive list of the assumptions of this approach.

These are fair points. We do not know of any other point-in-time disasters that would have occurred in our study period but cannot entirely rule out such contribution. We have added a caveat to the list of limitations at the top of p13.

To address the issue of sensitivity to the number of years included in the baseline, we have explored a shorter historical window of 4 years (2016-2020) instead of 6 years in the original analysis (2014-2020). The results are shown in Figure S9 and commented in the Results section (top of p 7) and discussion (limitations, top of p 13). Estimates of all-cause and respiratory excess mortality were extremely robust to the length of the calibration data for the model baseline, and so were all national cause-specific estimates. State- and cause-specific estimates were more sensitive to this choice, especially for Alzheimer’s and heart diseases, due to short-term trends and small counts in some of the time series. We note that our main conclusions are based on national estimates, which were less sensitive to the choice of calibration data.

3. There are no mentions in the description of mortality data of two common issues with vital registration data. First, the presence of ill-defined deaths (R chapter, and when subdividing injuries, Y10-Y34). If these increase or decrease during the pandemic because of differentials in coding, it may over or underestimate the presence of excess mortality. Second, the lack of coverage of all deaths. Helleringer talks a bit about that here https://academic.oup.com/ije/article/51/1/85/6460626?login=true in the context of estimating excess mortality. While this issue may seem trivial in the US, Murray shows here that there is a lack of complete coverage that is differential by county in the US https://journals.plos.org/plosmedicine/article?id=10.1371/journal.pmed.1000262. While this may or may not affect the authors' analysis, I think the implicit assumption of no changes in the coverage of deaths should be made explicit.

We added a mention of coverage issues and changes in reporting of ill-defined deaths to the limitation section (top of p13) and refer to the Murray paper. We note that ill-defined deaths would be captured in all-cause mortality but not in cause-specific mortality. Further, codes Y10-34 (unintentional intent injuries) are combined with the codes reflecting known intent injuries in the mortality datasets that we use here, so a large change in these codes would likely be captured in our analyses.

4. We are concerned about the analysis estimating the impact of NPIs. The authors are using explicit causal language throughout this section, but there's very little discussion about the methods used and the assumptions and limitations of these methods. Specifically:a. Please provide a bit more information on the model used for direct vs. indirect effects. We'd like to see explicit discussion of the assumptions and limitations of the approach but also of the stringency index used. Was the association between stringency index and excess deaths assumed to be linear? Or were different functional forms considered? It is also not clear how well the model fit the data.

Thank you. We have entirely revamped our analysis of direct and indirect effects and provided more information in the main text and supplement. A list of these changes is provided here:

We now use generalized additive models (gam) which allow for non-linear relationships between covariates and excess mortality. Indeed, we do find evidence for non-linear relationships with covariates, particularly with NPI.Our initial analysis was focused on predictors measuring COVID19 activity and the Oxford index, which is a broad indicator of the sum of NPIs in any community. We have now expanded the analysis to include hospital ICU occupancy in the model, as there could be a separate effect from COVID19 intensity and NPIs.We have also provided model fit statistics in the text and several figures to illustrate model projections vs observations (Figures 4 and S16-19).We have added caveats to the discussion p 13: “Though gam models indicate that the strength of interventions is a dominant predictor of excess mortality from external causes and all-cause mortality in young age groups, the relationships are non-linear and the resulting models do not fully capture mortality changes during the pandemic.”We have combed through the paper to remove the causal language and emphasized that the statistical relationships with interventions are not causal and further exploration of the mechanisms at play is needed.

b. Related to the above, please provide more details on how the results of the regressions were translated into the results presented. The main text reports percentages, but the methods only briefly explain how numbers of direct deaths were calculated, and the supplementary tables report coefficients. It is not clear if these estimates of direct and indirect deaths were somehow constrained to add up to the total number of excess deaths, but it doesn't seem like it since point estimates cross 100% in some cases.

That’s correct; estimates are not constrained to 100%. We have provided more details about the estimation of the percentages (=attributable fraction) in the main text (middle of p6) and in the supplement (p 4-5). Briefly, to estimate the direct impact of COVID19, we calculate expected excess deaths with the full gam model, where all predictors are set to their observed values, and with a zero-COVID19 model, where the COVID19 predictor is set to zero. The estimated fraction attributable to COVID19 (direct impact of the pandemic) is the difference between the predicted values by the full model and the zero-COVID19 model, divided by the predictions of the full model. The full equation is provided in supplement p4-5. Confidence intervals on these percentages are based on resampling excess mortality estimates, refitting the gam models, and repeating the above procedure 1000 times for each mortality stratum.

We think important to not restrict estimates to remain below 100% since we find consistent percentages above 100% in both gam approach and the “empirical” comparison with official statistics (Table 4, estimates for seniors). These results point at the possible effects of harvesting and/or inflated baseline due to depressed pathogen circulation.

c. Please be more explicit about causal assumptions (see comment 6 of reviewer 2).

We have clarified the attribution model in methods and supplement and removed causal language.

d. Please discuss the potential limitations of using the stringency index to quantify NPIs particularly since it doesn't take into account the actual implementation of the measures (and how this may have varied over time).

We have added several sentences to this effect in the caveats section p 13. “Generalized additive models indicate that a proxy for the strength of interventions is a dominant predictor of excess mortality from external causes and all-cause mortality in individuals under 45 yrs. Yet, the relationships are non-linear, and the resulting models do not fully capture mortality changes during the pandemic. Along the same lines, the Oxford stringency index used as a proxy for interventions does not consider the actual implementation nor the effect of interventions; it is solely based on mandates in place in different locations and time periods. We also assume that, for a given level of stringency, the impact of interventions does not change over time. Because time and intervention stringency are highly conflated, it would be difficult to study temporal variation in this relationship.”

Reviewer #2 (Recommendations for the authors):In this paper, the authors examine the impacts of the COVID-19 pandemic on excess mortality in the US up to April 30, 2021. The authors separate direct impacts (caused by COVID-19, coded as such or not) of the pandemic from indirect impacts (disruptions), finding that most excess deaths (90%) are due to direct impacts. Importantly, the authors find that the official COVID-19 death tally is an undercount of these deaths. Moreover, the authors also find that excess deaths due to other causes are the main driver of excess mortality among younger populations. This is a very important research question that the authors have addressed using robust methods. I have a few suggestions that I hope can help clarify the reporting of results and outlining of assumptions.1. Abstract: since the study presents several axes of comparison (old vs young, direct vs indirect, specific causes vs one another) the abstract is quite complicated, and there are a few sections that may benefit from clarification. Specifically: (a) the authors mention the indirect effects

It looks like the end of this comment may have been cut; but the point about the abstract being complicated is well taken. We have hopefully simplified the abstract.

2. The framing of the study in the introduction may benefit from a more comprehensive outlining of potential directions that mortality may change with the pandemic. For example, declines in mortality have been observed in many settings (e.g. Chile and Guatemala; or in Figure 3 of this paper) following lockdowns in areas with low COVID-19 spread. This is mostly due to declines in external causes (mostly traffic crashes) and air pollution-related deaths (e.g. CVD or infant deaths). The authors just mention one potential directionality (COVID-19 increasing some deaths directly, and NPIs causing other deaths) without any mention to a lowering of mortality that may obscure these increases.

Thanks. We do mention the potential mortality declines brought about by lockdown in discussion (end of p13). In the revised introduction (p3), we have provided a fuller list of mechanisms that may increase or decrease mortality during the pandemic and explained how we would expect to see these mechanisms manifest in data.

3. Choosing the baseline/counterfactual is a key part of excess mortality analyses. I've seen other papers exploring different potential periods (shorter periods; either by using older or newer years, e,g 2017-2020 or 2014-2017) to test sensitivity to the choice of these periods. Moreover, I would think that an absence of other point-in-time disasters would also be a needed assumption (as these would not represent a baseline). It may be beneficial for the reader to outline explicitly the assumptions of their model. Some of these assumptions are outlined on page 5, but I am not sure whether this is an extensive list of the assumptions of this approach.

The reviewer makes a good point that the choice of historic years used for calibration of the model baseline is important. We do not know of any other point-in-time disasters that would have occurred in our study period.

To address the issue of sensitivity to the number of years included in baseline, we have tried a shorter historical window of ~4 years (2016-2020) instead of ~6 years used in the original analysis (2014-2020). The results are shown in Figure S9 and commented in the Results section (top of p 7) and discussion (limitations, bottom of p 12). Estimates of all-cause and respiratory excess mortality were extremely robust to the length of the calibration data used for the model baseline, as well as national estimates for any cause. State- and cause-specific estimates were more sensitive to this choice, especially for Alzheimer’s and heart diseases, due to short-term trends and small counts in some of the time series. We note that our main conclusions are based on national estimates, which were less sensitive to the choice of calibration data.

4. I see no mentions in the description of mortality data of two common issues with vital registration data. First, the presence of ill-defined deaths (R chapter, and when subdividing injuries, Y10-Y34). If these increase or decrease during the pandemic because of differentials in coding, it may over or underestimate the presence of excess mortality. Second, the lack of coverage of all deaths. Helleringer talks a bit about that here https://academic.oup.com/ije/article/51/1/85/6460626?login=true in the context of estimating excess mortality. While this issue may seem trivial in the US, Murray shows here that there is a lack of complete coverage that is differential by county in the US https://journals.plos.org/plosmedicine/article?id=10.1371/journal.pmed.1000262. While this may or may not affect the authors' analysis, I think the implicit assumption of no changes in the coverage of deaths should be made explicit.

We added a mention of coverage issues and changes in reporting of ill-defined deaths to the limitation section and refer to the Murray paper. We note that ill-defined deaths would be captured in all-cause mortality but not in cause-specific mortality. Further, codes Y10-34 (unintentional intent injuries) are combined with the codes reflecting known intent injuries in the real time mortality data, so a large change in these codes would probably be captured in our analyses.

5. Direct deaths: the authors use COVID-19, influenza, pneumonia, and COPD deaths as an indicator of respiratory mortality. Is this the list of ICD codes in P28 (2.a) of the appendix? The authors also use "mention of COVID-19" as an indicator of directly caused deaths. How is this operationalized? Is it ICD10 code U07? Or does this not rely on ICD coding? The list of respiratory codes in the supplement makes it seem like this is ICD based, but it is unclear from the text itself.

The list of ICD codes is provided in supplement p1. Our entire analysis is based on ICD codes, and we use U07 for COVID-19. We have clarified this in methods of the main text (p4). Note that throughout the paper, we use the aggregated time series made available in near real time throughout the pandemic by CDC/NCHS, so the data have already been tabulated by select categories of codes and we cannot create different categories.

6. I am concerned about the analysis estimating the impact of NPIs. The authors are using explicit causal language throughout this section, but there's very little discussion about the causal assumptions underpinning this analysis. Specifically: (a) I see no approach to addressing lack of exchangeability (for example, areas that implemented more NPIs may have a more vulnerable population, hence the political will to implement them); (b) no discussion of issues with levels of analysis from a jurisdictional point of view, since the level of devolvement of power from states to local jurisdictions varies widely by state, making state-level analysis potentially tricky (and county-level ones too for that matter); and (c) issues with measurement error, specifically differential measurement error of the outcome, influenced by the exposure (improved vital registration coding in areas with more or less NPIs due to better established public health systems). Last, the authors are framing this analysis as a way to "quantify the direct and indirect impacts of the pandemic on different causes of death". I am failing to grasp the intuition of how this analysis helps with this objective, so it may need to be explicitly set in the methods section.

We have expanded our list of caveats for this analysis based on the comments by this and other reviewers (p 13, the full section is also appended below). In particular, we have added comments regarding differential vulnerability of the populations, changes in coding and coverage of vital statistics during COVID-19, and lack of consideration of NPI implementation at the local level.

Limitation section p 13: “Our study is subject to several limitations. First, mortality counts below the minimum cut-off value of 10 were suppressed due to privacy regulations. As a result, our age-specific analyses were restricted to larger states, and we could not assess the role of race/ethnicity. Prior work has shown important disparities in COVID-19 impact by race/ethnicity and economic status (Mena et al., 2021; Rossen, 2021) in the US and abroad. Second, official coding practices may have changed between states and through time based on SARS-CoV-2 testing availability, location of death, demographic factors, and comorbidities. Third, we assumed full coverage of deaths reporting, which may not be valid throughout the US (Murray et al., 2010), and we did not study changes in deaths ascribed to ill-defined codes (R codes). Ill-defined deaths would be captured in all-cause mortality but not in cause-specific analyses. Fourth, we find periods of negative excesses in cancer (throughout the pandemic), cardiovascular, and heart diseases, possibly due to changes in ascertainment of underlying cause of death (e.g. a death in a cancer patient with COVID-19 is ascribed to COVID-19), harvesting (Saha et al., 2013), or depressed circulation of endemic pathogens other than influenza. Fifth, we choose to fit the model baseline to data for 2014-2020, which is arbitrary. We studied the sensitivity of our excess mortality estimates to this assumption (Supplement). While national analyses were robust to this choice, as well as state-level analyses of most conditions, state-specific estimates for Alzheimer’s disease were more sensitive. Further, the model baseline does not account for possible point-in-time disasters that may have occurred during the pandemic but are independent of COVID19 (eg, a hurricane) or changes in air pollution. Sixth, generalized additive models indicate that the Oxford stringency index, a proxy for the strength of interventions, is a dominant predictor of excess mortality from external causes and all-cause mortality in individuals under 45 yrs. Yet, the relationships are non-linear, and the resulting models do not fully capture mortality changes during the pandemic. Along the same lines, the Oxford stringency index does not consider the actual implementation nor the effect of interventions; it is solely based on mandates in place in different locations and time periods. We also assume that, for a given level of stringency, the impact of interventions does not change over time. Because time and intervention stringency are highly conflated, it would be difficult to study potential temporal variation in this relationship. Further, analyses are aggregated at the state or national level, while implementation of interventions may operate more locally. We also do not account for underlying differences in vulnerability between states, where more vulnerable states may have implemented stricter interventions (although this potential bias would not affect temporal analyses). Finally, our study ends on January 1, 2022 and does not capture a recrudescence of COVID-19-related deaths due to the Omicron variant. As a result, our excess mortality estimates should be deemed conservative.”

7. I want to commend the authors for their presentation of results. Specifically, the rounding of mortality estimates to avoid false notions of precision and very clear figures. A few notes on figures: (a) figure 1 is lacking a title in my PDF, and the black line is not included in the legend (I am guessing it corresponds to the state in the facet title); (b) figure 2 is missing confidence intervals for respiratory illnesses; (c) I suggest the authors map the results of table 1, as a spatial pattern will most likely emerge (top by excess mortality are all Deep South or Northeast states).

Thank you. We have added a title to figure 1 and a legend for the back line. As regards mapping all-cause excess mortality, now that every state has experienced multiple waves of COVID19, and each state was hit hard at a different time, the geography of excess mortality for the combined study period of March 1, 2020 to January 1, 2022 is not as meaningful.

8. The results comparing COVID-19 to the 2017/2018 flu season are very interesting, as they help contextualize the impact. However, I am a bit concerned about "denominators" here, specifically time. The authors compared 13 months of excess mortality to an undetermined number of months (4? 6? 12?). I am not sure how influenza seasons are conceptualized, and I think it'd be good to be explicit with its duration. I understand that the authors may be wanting to compare a "full season" of influenza vs one of COVID-19, but is this direct comparison even possible, considering the different (or lack of?) seasonality of COVID-19?

The reviewer is correct that these comparisons were for different epidemic durations. We have revamped this section. It is tricky to find a particular time period that is relevant for comparison since COVID19 mortality has been quite heterogeneous across the US. Yet we elected to use the November to March period for both pathogens, comparing the severe A/H3N2 flu season in Nov 2017-Mar 2018, with the first winter wave of COVID19 from Nov 2020-Mar 2021 (where excess mortality was experienced by all states). This comparison highlights the magnitude of COVID-19, where excess mortality is 5-6 fold higher than excess mortality from flu, just for these 5 months. The relevant sections of results (middle of p 7) and Appendix have been updated and a new figure was added to supplement (Figure S11).

9. There's another result I am having trouble (a) interpreting it and (b) understanding where it comes from: % of deaths directly attributable to COVID-19. The authors seem to be interpreting this as direct effects of COVID-19 infection, as opposed to indirect effects due to disruptions, at least according to the appendix. My comment 2 above already covers the idea that only thinking about increases in mortality due to NPIs is an incomplete framing, as there are reasons why we expect deaths to actually decline during lockdowns (traffic or air pollution-related ones). There's a cursory mention of this in the appendix, but it is restricted to other respiratory deaths. To follow the authors' example in the appendix, a person with diabetes may also not have died because the air pollution levels in their city plummeted and the MI they were about to have did not happen. Or because their car didn't crash (or another car didn't crash on them) because traffic levels plummeted too. On (b), understanding where it comes from, the appendix provides extra clues (the main text doesn't explain this part that much). If I am understanding things right, the authors are regressing age and cause-specific mortality on NPI strength and on weekly deaths. They are then multiplying the COVID-19 deaths coefficient by the number of deaths to get the # of deaths directly caused by COVID-19. The authors do test potential lags, as both NPIs and deaths may have lagged effects or may be the result of lags between infections and deaths. However, I don't fully follow: (a) why the NPI adjustment is needed at all, (b) why are deaths an adequate COVID-19 activity indicator, especially since authors find a 0-week lag (pretty self-fulfilling, given that they are regressing deaths on a subset of deaths), which doesn't speak much about healthcare disruptions (which occur weeks before that point).

We do not consider NPI as an adjustment, merely as a potential covariate representing a broad set of interventions that may affect mortality. As the reviewer points out, an effect of interventions on mortality could be either positive or negative, for instance lockdown reducing pollution in turn reducing cardiac deaths, or lockdown increasing mental health issues in turn increasing suicide or opioids deaths. We have now included a fuller list of mechanisms involving interventions in the introduction (including air pollution) to clarify our thinking and justify the choice of the model with COVID indicator and NPI (p3 of introduction). As regards healthcare disruptions, we have expanded the attribution model to include ICU occupancy.

Our approach for estimating the % of deaths directly due to COVID19 has changed as well. We use a gam model to identify the relationship between excess deaths and covariates (COVID19, NPI and ICU), and compare the excess mortality predictions of the full gam model with COVID19, NPI and ICU covariates to their observed values, vs those of the same model but where COVID19 is set to zero. This is now detailed in methods and supplement.

One last concern I have is of competing risks, and whether areas with more strict NPIs would have lower direct excess mortality, increasing the % of deaths that are indirectly attributable to the pandemic, not because these deaths were caused by the pandemic, but because they could not have been caused by a well-controlled epidemic (given NPIs).

We try to address this point by discussing countries which have experienced lockdowns without a COVID19 epidemic (eg Australia before Omicron, Russia before the wild-type variant arrived, end p 13). In essence, if NPIs work to control SARS circulation, then mortality should either be at baseline levels (hence excess mortality is zero), or below baseline levels (negative excess mortality). Australia and Russia seem to have experienced the former (zero excess mortality).